# Impact of Siberian observations on the optimization of surface $CO_2$ flux

**Jinwoong Kim[1]\*, Hyun Mee Kim[1], Chun-Ho Cho[2], Kyung-On Boo[2], Andrew R. Jacobson[3, 4], Motoki Sasakawa[5], Toshinobu Machida[5], Mikhail Arshinov[6], and Nikolay Fedoseev[7]**

[1]{Department of Atmospheric Sciences, Yonsei University, Seoul, Republic of Korea}

[2]{National Institute of Meteorological Sciences, Jeju, Republic of Korea}

[3]{Earth System Research Laboratory, National Oceanic and Atmospheric Administration, Boulder, USA}

[4]{Cooperative Institute for Research in Environmental Sciences, University of Colorado, Boulder, USA}

[5]{Center for Global Environmental Research, National Institute for Environment Studies, Tsukuba, Japan}

[6]{V. E. Zuev Institute of Atmospheric Optics, Russian Academy of Sciences, Tomsk, Russia}

[7]{Melnikov Permafrost Institute, Russian Academy of Sciences, Yakutsk, Russia}

Correspondence to: Hyun Mee Kim (khm@yonsei.ac.kr)

\*Current affiliation: Climate Research Division, Environment and Climate Change Canada, Toronto, Canada

**Abstract**

To investigate the effect of additional $CO_2$ observations in the Siberia region on the Asian and global surface $CO_2$ flux analyses, two experiments using different observation datasets were performed for 2000-2009. One experiment was conducted using a data set that includes additional observations of Siberian tower measurements (Japan-Russia Siberian Tall Tower Inland Observation Network: JR-STATION), and the other experiment was conducted using a data set without the above additional observations. The results show that the global balance of

the sources and sinks of surface $CO_2$ fluxes was maintained for both experiments with and without the additional observations. While the magnitude of the optimized surface $CO_2$ flux uptake and flux uncertainty in Siberia decreased from -1.17±0.93 Pg C yr$^{-1}$ to -0.77±0.70 Pg C yr$^{-1}$, the magnitude of the optimized surface $CO_2$ flux uptake in the other regions (e.g., Europe) of the Northern Hemisphere (NH) land increased for the experiment with the additional observations, which affect the longitudinal distribution of the total NH sinks. This change was mostly caused by changes in the magnitudes of surface $CO_2$ flux in June and July. The observation impact measured by uncertainty reduction and self-sensitivity tests shows that additional observations provide useful information on the estimated surface $CO_2$ flux. The average uncertainty reduction of the Conifer Forest of EB is 29.1% and the average self-sensitivities at the JR-STATION sites are approximately 60% larger than those at the towers in North America. It is expected that the Siberian observations play an important role in estimating surface $CO_2$ flux in the NH land (e.g., Siberia and Europe) in the future.

## 1   Introduction

The terrestrial ecosystem in the Northern Hemisphere (NH) plays an important role in the global carbon balance (Hayes et al., 2011; Le Quéré et al., 2015). Especially, Siberia is considered to be the one of the largest $CO_2$ uptake regions and reservoirs due to its forest area (Schulze et al., 1999; Houghton et al., 2007; Tarnocai et al., 2009; Kurganova et al., 2010; Schepaschenko et al., 2011) and its dynamics and interactions with the climate have global significance (Quegan et al., 2011). Therefore, it is important to accurately estimate the surface $CO_2$ fluxes in this region. For instance, Dolman et al. (2012) estimated terrestrial carbon budget of Russia, Ukraine, Belarus, and Kazakhstan using inventory-based, eddy covariance, and inversion methods and showed that the carbon budgets produced by three methods agree within their uncertainty bounds.

To estimate the surface $CO_2$ flux, atmospheric $CO_2$ inversion studies are conducted using atmospheric transport models and atmospheric $CO_2$ observations (Gurney et al., 2002; Peylin et al., 2013). However, prior emission, measurement error of observation, observation operator including model transport, and representative error affect the uncertainty of atmospheric inversion results (Engelen et al., 2002; Berchet et al., 2015a). Along these factors, large uncertainties remain in the estimated surface $CO_2$ fluxes due to the sparseness of current

surface $CO_2$ measurements assimilated by inverse models (Peters et al., 2010; Bruhwiler et al., 2011). Peylin et al. (2013) performed an intercomparison study of estimated surface $CO_2$ fluxes from 11 different inversion systems. The results showed that the estimated surface $CO_2$ flux uptake in the NH, where the atmospheric $CO_2$ network is dense, is similar across the inversion systems; meanwhile, the established flux is noticeably different across the inversion systems for the tropics and the Southern Hemisphere (SH), where the atmospheric $CO_2$ network is sparse.

Regionally, however, the longitudinal breakdown of all the NH sinks appears to be much more variable than the total flux itself. Therefore, additional observations in a sparse $CO_2$ observation network region are necessary to reduce uncertainty in estimating the surface $CO_2$ flux. Maksyutov et al. (2003) showed that additional observations in the Asia region show the largest effect and reduce the uncertainty in the estimated regional $CO_2$ fluxes for Siberia during 1992-1996 by time-independent synthesis inversion. Chevallier et al. (2010) also argued that an extension of the observation network toward Eastern Europe and Siberia is necessary to reduce uncertainty in estimated fluxes by inversion methods. Despite the necessity of additional observations in this region, only a few atmospheric $CO_2$ inversion studies have been conducted using observations in this region due to the deficiency of observations (Quegan et al., 2011).

Meanwhile, Reuter et al. (2014) and Feng et al. (2016) reported that the European terrestrial $CO_2$ uptake inferred by the satellite-retrieved dry-air column-average mole fraction of $CO_2$ (XCO$_2$) is larger than that inferred by a bottom-up inventory approach or inverse modeling systems using surface-based $CO_2$ atmospheric concentrations. Though a broad spatial coverage of XCO$_2$ from satellite radiance observations provides useful information for inversion systems in quantifying surface $CO_2$ fluxes at various scales which is not provided by ground-based measurements, the current XCO$_2$ has low accuracy and regional biases of a few tenths of a ppm (parts per million), which may hamper the accuracy of estimated surface $CO_2$ fluxes (Miller et al., 2007; Chevallier et al., 2007). Therefore, in situ observations determined by surface measurements are necessary to more accurately estimate the surface $CO_2$ flux in the inverse models.

To supply additional observations over Siberia to inverse modeling studies, several efforts to observe the atmospheric $CO_2$ concentrations in Siberia have been conducted. For example, the Max Planck Institute (MPI) operates a tower (since April 2009), preceded by aircraft

measurements (from 1998 to 2005 with 12 to 21 day intervals) at Zotino (ZOTTO; 60.75°N, 89.38°E) (Lloyd et al., 2002; Winderlich et al. 2010). In addition, the Airborne Extensive Regional Observations in Siberia (YAK-AEROSIB) aircraft campaign in 2006 (Paris et al., 2008) and Trans-Siberian Observation Into the Chemistry of the Atmosphere (TROICA) project (Turnbull et al., 2009) have measured $CO_2$ and other chemical species. However, except Zotino that has multi-year measurements, these data collected during specific seasons or over only a few years do not provide the long-term $CO_2$ concentration data necessary to be used as a constraint in the inverse modeling system.

The Center for Global Environmental Research (CGER) of the National Institute for Environmental Studies (NIES) of Japan with the cooperation of the Russian Academy of Science (RAS) constructed a tower network called the Japan-Russia Siberian Tall Tower Inland Observation Network (JR-STATION) in 2002 to measure the continuous $CO_2$ and $CH_4$ concentrations (eight towers in central Siberia and one tower in eastern Siberia) (Sasakawa et al., 2010, 2013). The vertical profile of $CO_2$ concentrations from the planetary boundary layer (PBL) to the lower free troposphere is also measured by aircraft at one site of the JR-STATION sites (Sasakawa et al., 2010, 2013). Saeki et al. (2013) estimated the monthly surface $CO_2$ flux for 68 subcontinental regions by using the fixed-lag Kalman smoother and NIES-TM transport model with JR-STATION data. They reported that the inclusion of additional Siberian observation data has an impact on the inversion results showing larger interannual variability over northeastern Europe as well as Siberia, and reduces the uncertainty of surface $CO_2$ uptake. Meanwhile, Berchet et al. (2015b) estimated regional $CH_4$ fluxes over Siberia in 2010 by using JR-STATION data.

CarbonTracker, developed by the National Oceanic and Atmospheric Administration Earth System Research Laboratory (NOAA ESRL) (Peters et al., 2007), is an atmospheric $CO_2$ inverse modeling system that estimates optimized weekly surface $CO_2$ flux on a 1°×1° horizontal resolution by using the Ensemble Kalman Filter (EnKF) (Evensen 1994). Since the original CarbonTracker release (Peters et al 2007), a series of improvements have been made with subsequent releases. These include increasing the number of sites from which $CO_2$ data are assimilated, increasing the resolution of atmospheric transport, improving the simulation of atmospheric convection in TM5 (Krol et al., 2005) which is the transport model used in CarbonTracker, and the use of multiple first-guess flux models to estimate sensitivity to priors. These improvements are documented at http://carbontracker.noaa.gov. Several studies have

focused on Asia using CarbonTracker (Kim et al., 2012, 2014a, b; Zhang et al., 2014a, b). Schneising et al. (2011) showed that SCanning Imaging Absorption spectroMeter for Atmospheric CHartographY (SCIAMACHY) retrieval data indicate a stronger North American boreal forest uptake and weaker Russian boreal forest uptake compared to CarbonTracker within their uncertainties. On the other hand, Zhang et al. (2014b) estimated surface $CO_2$ fluxes in Asia by assimilating CONTRAIL (Machida et al., 2008) aircraft $CO_2$ measurements into the CarbonTracker framework. The CONTRAIL measurements include ascending/descending vertical profiles and cruise data below tropopause. The results show that surface $CO_2$ uptake over the Eurasian Boreal (EB) region slightly increases from -0.96 Pg C yr$^{-1}$ to -1.02 Pg C yr$^{-1}$ for the period 2006-2010 when aircraft $CO_2$ measurements were assimilated. However, the surface measurements data over the EB region are still not used in the study by Zhang et al. (2014b). Using an influence matrix calculation, Kim et al. (2014b) showed that comprehensive coverage of additional observations in an observation sparse region, e.g., Siberia, is necessary to estimate the surface $CO_2$ flux in these areas as accurately as that obtained for North America in the CarbonTracker framework.

In this study, the impact of additional Siberian observations on the optimized surface $CO_2$ flux over the globe and Asian region within CarbonTracker (The version of CarbonTracker used in this study is based on the CarbonTracker 2010 release.) is investigated by comparing the results of estimated surface $CO_2$ fluxes from two experiments with and without Siberian observations. Section 2 presents the methodology including a priori flux data, atmospheric $CO_2$ observations, and experimental framework. Section 3 presents the results, and Section 4 provides a summary and conclusions.

## 2   Methodology

### 2.1   Inversion method

CarbonTracker is an inverse modeling system developed by Peters et al. (2007). Optimized surface $CO_2$ fluxes with a 1°×1° horizontal resolution are calculated as follows:

$$F(x,y,t) = \lambda_r \cdot F_{bio}(x,y,t) + \lambda_r \cdot F_{ocn}(x,y,t) + F_{ff}(x,y,t) + F_{fire}(x,y,t), \qquad (1)$$

where $F_{bio}(x,y,t)$, $F_{ocn}(x,y,t)$, $F_{ff}(x,y,t)$, and $F_{fire}(x,y,t)$ are a priori emissions from the biosphere, the ocean, fossil fuel, and fires. $\lambda_r$ is the scaling factor to be optimized in the data

assimilation process, corresponding to 156 regions around the globe (126 land and 30 ocean regions). In the land, the ecoregions are defined as the combination of 11 land region of Transcom regions (Gurney et al., 2002) with 19 land-surface characterization based on Olson et al. (1992). Inappropriate combinations of TransCom regions and Olson types are excluded. In the ocean, 30 ocean regions are defined following Jacobson et al. (2007). The scaling factor spans 5 weeks with 1 week resolution. Several previous studies for CarbonTracker (e.g., Peters et al., 2007; 2010, Kim et al., 2012, 2014a, b; Zhang et al., 2014a, b; van der Laan-Luijkx et al., 2015) showed that 5 weeks of lag and 1-week time resolution are appropriate for optimizing the surface $CO_2$ fluxes. In each assimilation cycle (i.e., analysis step), the entire scaling factor for 5 weeks is updated by 1 week observations measured in the most recent week by a time stepping approach. The smoother window moves forward by 1 week at each assimilation cycle. After 5 assimilation cycles, the first part of the scaling factor analyzed by 5 weeks observations is regarded as the optimized scaling factor. More detailed information of the assimilation process can be found in Kim et al. (2014b).

The ensemble Kalman filter (EnKF) data assimilation method used in CarbonTracker is the ensemble square root filter (EnSRF) suggested by Whitaker and Hamill (2002). The analysis equation for data assimilation is expressed as

$$x^a = \mathbf{K}y^o + (\mathbf{I}_n - \mathbf{KH})x^b, \tag{2}$$

where $x^a$ is the n-dimensional analysis (posterior) state vector ; $y^o$ is the p-dimensional observation vector (atmospheric $CO_2$ observations); $\mathbf{K}$ is the n × p dimensional Kalman gain; $\mathbf{I}_n$ is the identity matrix; $\mathbf{H}$ is the linearized observation operator, which transforms the information in the model space to the information in the observation space; and $x^b$ is the background state vector. In CarbonTracker, the state vector corresponds to the scaling factor. The Kalman gain $\mathbf{K}$ is defined as

$$\mathbf{K} = \left(\mathbf{P}^b\mathbf{H}^T\right)\left(\mathbf{HP}^b\mathbf{H}^T + \mathbf{R}\right)^{-1}, \tag{3}$$

where $\mathbf{P}^b$ is the background error covariance; $\mathbf{R}$ is the observation error covariance or model data mismatch, which is predefined at each observation site. $\mathbf{P}^b\mathbf{H}^T$ and $\mathbf{HP}^b\mathbf{H}^T$ in Eq. (3) can be calculated as

$$\mathbf{PH}^T \approx \frac{1}{m-1}\left(x_1', x_2', \ldots, x_m'\right) \cdot \left(\mathbf{H}x_1', \mathbf{H}x_2', \ldots, \mathbf{H}x_m'\right)^T, \tag{4}$$

$$\mathbf{HPH}^{\mathrm{T}} \approx \frac{1}{m-1}\left(\mathbf{Hx}_1', \mathbf{Hx}_2', \ldots, \mathbf{Hx}_m'\right) \cdot \left(\mathbf{Hx}_1', \mathbf{Hx}_2', \ldots, \mathbf{Hx}_m'\right)^{\mathrm{T}},$$ (5)
where $m$ is the number of ensembles and $'$ denotes the perturbation of ensemble mean.
The sampling error caused by the limited ensemble size may degrade the analysis accuracy.
To reduce the impact of sampling error in the EnKF, the covariance localization method is
used (Houtekamer and Mitchell, 2001). The localization is not applied to Marine Boundary
Layer (MBL) sites, e.g., observation sites in Antarctica, because the MBL sites are considered
as including information on large footprints of flux signals (Peters et al., 2007). The physical
distance between the scaling factors cannot be defined. Therefore, localization is performed
based on the linear correlation coefficient between the ensemble of the scaling factor and the
ensemble of the model $CO_2$ concentration (Peters et al., 2007). A statistical significance test is
performed on the linear correlation coefficient with a cut-off at a 95% significance in a
student's T-test. Then the components of Kalman gain with an insignificant statistical value
are set to zero.
After one analysis step is completed, the new mean scaling factor that serves as the
background scaling factor for next analysis cycle is predicted as
$$\lambda_t^b = \frac{(\lambda_{t-2}^a + \lambda_{t-1}^a + 1)}{3},$$ (6)
where $\lambda_t^b$ is a prior mean scaling factor of the current analysis cycle, $\lambda_{t-2}^a$ and $\lambda_{t-1}^a$ are
posterior mean scaling factors of previous cycles. Eq. (6) propagates information from one
step to the next step (Peters et al., 2007).
The detailed algorithm of inversion method used in this study can be found in Peters et al.
(2007) and Kim et al. (2014a).
**2.2  A priori flux data**
Four types of a priori and imposed $CO_2$ fluxes used in this study are as follows: (1) First guess
biosphere flux from the Carnegie–Ames–Stanford Approach Global Fire Emissions Database
(CASA GFED) version 3.1 (van der Werf et al., 2010). The 3 hour interval Net Ecosytem
Exchange (NEE) is calculated from monthly mean Net Primary Production (NPP) and

ecosystem respiration (RE) by using a simple temperature $Q_{10}$[1] relationship and a linear scaling of photosynthesis with solar radiation (Olsen and Randerson, 2004); (2) the prior ocean flux from air-sea partial pressure differences based on Jacobson et al. (2007). Short-term flux variability is derived from the atmospheric model wind speeds via the gas transfer coefficient; (3) biomass burning emissions obtained from GFED v3.1 (van der Werf et al., 2010); (4) the prescribed fossil fuel emission from the Carbon Dioxide Information and Analysis Center (CDIAC, Boden et al., 2010) and the Emission Database for Global Atmospheric Research (EDGAR, European Commission, 2009) databases. The annual global total fossil fuel emissions are based on CDIAC. Fluxes at 1°x1° resolution are spatially distributed according to the EDGAR inventories.

## 2.3  Atmospheric CO₂ observations

Atmospheric $CO_2$ mole fraction observations measured at surface observation sites are used in this study. Figure 1 shows the observation network and Table 1 presents observation site information for the Asian and European regions. Three sets of atmospheric $CO_2$ observations data are assimilated: (1) surface $CO_2$ observations distributed by the NOAA ESRL (observation sites operated by NOAA, Environment Canada (EC), the Australian Commonwealth Scientific and Industrial Research Organization (CSIRO), the National Center for Atmospheric Research (NCAR), and Lawrence Berkeley National Laboratory (LBNL)) (observation data is available at http://www.esrl.noaa.gov/gmd/ccgg/obspack/ data.php; Masarie et al., 2014); (2) World Data Centre for Greenhouse Gases (WDCGG, http://ds.data.jma.go.jp/wdcgg/); (3) JR-STATION observation data over Siberia operated by CGER/NIES (Sasakawa et al., 2010, 2013). The JR-STATION sites consist of nine towers (eight towers in west Siberia and one tower in east Siberia). Atmospheric air was sampled at four levels on the BRZ tower and at two levels on the other eight towers. At the BRZ (Berezorechka) site in west Siberia, both tower and aircraft measurements are sampled. The light aircraft at BRZ site measures the vertical profiles of $CO_2$ from the PBL to the lower free troposphere and these vertical profiles are used as independent observations for verification.

---

[1] It is calculated as $Q_{10}(t) = 1.5^{((T_{2m}-T_0)/10.0)}$, where $t$ is time, $T_{2m}$ is temperature (K) at 2 m, and $T_0$ is 273.15 K.

Sampled $CO_2$ data were calibrated against the NIES 09 $CO_2$ scale which is lower than the
WMO-X2007 $CO_2$ scale by 0.07 ppm at around 360 ppm and consistent in the range between
380 and 400 ppm (Machida et al., 2011). Detailed description of JR-STATION sites can be
found in Sasakawa et al. (2010, 2013). Daytime averaged $CO_2$ concentrations (1200-1600
LST, representing the time when active vertical mixing occurred in the PBL) for each day
from the time series at the highest level of tower measurements are used in the data
assimilation.
In CarbonTracker, model data mismatch (MDM, $\mathbf{R}$ in Eq. (7)) is assigned by site categories.
The location of each observation site is represented in Fig. 1. The assigned MDM requires
innovation $\chi^2$ statistics in Eq. (7) become close to one at each observation site (Peters et al.
11 2007).

$$\chi^2 = \frac{(y^o - \mathbf{H}x^b)^2}{\mathbf{H}\mathbf{P}^b\mathbf{H}^T + \mathbf{R}},  \tag{7}$$
where $y^o - \mathbf{H}x^b$ represent the innovation. The site categories and MDM values are assigned the
same value as in previous studies (Peters et al., 2007; Kim et al. 2014b; Zhang et al., 2014b):
marine boundary layer (0.75 ppm), continental sites (2.5 ppm), mixed land/ocean and
mountain sites (1.5 ppm), continuous sites (3.0 ppm), and difficult sites (7.5 ppm) that are
located near polluted areas with high anthropogenic $CO_2$ emissions. Continuous site category
is generally used for observations measured continuously. For the JR-STATION sites that
have continuous tower measurements, the MDM is set to 3 ppm, which is the same as tower
measurements in North America.
**2.4 Experimental framework**
Two experiments with different set of observations are conducted in this study: one
experiment, the CNTL experiment, is conducted by using set of observations without
observations in the Siberia region (black color observation sites represented in Fig. 1); the
other experiment, the JR experiment, is conducted using all available observations including
the Siberia data (all observation sites represented in Fig. 1). The TM5 model runs at global
3°×2° horizontal resolution and a nesting domain centered in Asia with 1°×1° horizontal
resolution. The nesting domain is shown in Fig. 1. Meteorological variables for running the
TM5 transport model are from the European Centre for Medium-Range Weather Forecasts
(ECMWF) forecast model output. The experimental period is from 1 January 2000 to 31

December 2009. The observation data commonly used for the CNTL and JR experiments exist from 2000, but the additional Siberia data for the JR experiment exist from 2002. The number of ensembles is 150, and the scaling factor includes 5 weeks of lag, as in previous studies (Peters et al., 2007, 2010; Peylin et al., 2013; Kim et al., 2012, 2014a b; Zhang et al., 2014a, b).

## 3 Results

### 3.1 Characteristics of carbon fluxes

In this section, optimized surface $CO_2$ fluxes inferred from the two experiments are examined. The optimized surface $CO_2$ flux in 2000 and 2001 is excluded from this analysis because 2000 is considered a spin-up year similar to previous studies using CarbonTracker, and JR-STATION data are used since 2002. Only the biosphere and ocean fluxes are presented here because fires (biomass burning) and fossil fuel emissions are not optimized in CarbonTracker.

Figure 2 presents the spatial distribution of the averaged prior and optimized biosphere and ocean fluxes of the two experiments and the difference between the CNTL and JR experiments from 2002 to 2009. The optimized biosphere flux uptakes of the CNTL and JR experiments are globally 1.60 ~ 1.61 Pg C yr$^{-1}$ greater than the prior flux uptakes (Figs. 2a, c, d, Table 2). The difference in fluxes between the prior and JR experiment is large in EB (Figs. 2a, d) although smaller than that between the prior and CNTL experiment (Figs, 2a, c). The differences in fluxes between the CNTL and JR experiments are large in EB (Siberia) where the new additional observations are assimilated (Fig. 2b). The magnitude of surface $CO_2$ uptakes decreases in that region by assimilating JR-STATION observation data. On the contrary, the average surface $CO_2$ uptakes in other regions, such as North America, Europe, the western North Pacific Ocean, and the Atlantic Ocean, increase by assimilating JR-STATION observation data.

The difference in the optimized $CO_2$ flux between the two experiments is analyzed. Table 2 presents prior and optimized fluxes with their uncertainties for global total, global land, global ocean, NH total, Tropics total, Southern Hemisphere total, and TransCom regions in the NH. Flux uncertainties are calculated from the ensembles of prior and optimized surface fluxes assuming Gaussian errors, following previous method used in Peters et al. (2007, 2010). The global total biogenic and oceanic optimized $CO_2$ fluxes are similar for the CNTL experiment

$(-5.54\pm1.85$ Pg C yr$^{-1}$) and JR experiment $(-5.55\pm1.72$ Pg C yr$^{-1}$), compared with the global
prior flux of $-3.94\pm2.24$ Pg C yr$^{-1}$. The global land sink in the CNTL experiment is larger by
0.07 Pg C yr$^{-1}$ than that of the JR experiment, and the global ocean sink in the CNTL
experiment is smaller by 0.08 Pg C yr$^{-1}$ than that of the JR experiment. The additional
observations do not introduce any discrepancy between the two experiments with respect to
the global total sink, and they indicate only a small difference in the land-ocean $CO_2$ flux
partitioning. The estimated $CO_2$ flux uncertainty in the land region from the JR experiment is
smaller than that of the CNTL experiment because new observations provide additional
constraints on the optimized $CO_2$ flux. For specific regions in the NH, a large difference of
optimized surface $CO_2$ flux between the CNTL and JR experiments is observed in the EB.
The largest increment between a priori and CNTL is shown in EB where the least in situ
observations are available as shown in Fig. 1. The other regions where more local
observations are available show smaller increments. The surface $CO_2$ uptake in the EB of the
CNTL experiment is $-1.17\pm0.93$ Pg C yr$^{-1}$ and that of the JR experiment is $-0.77\pm0.70$ Pg C
yr$^{-1}$, respectively. As expected, the uncertainty of the optimized surface $CO_2$ uptake in the EB
in the JR experiment is reduced by assimilating additional observations. In contrast, the
surface $CO_2$ uptake increases in other regions of the NH where no additional observations are
assimilated.
Figure 3 presents the spatial distribution of the optimized biosphere fluxes difference between
the CNTL and JR experiments from 2002 to 2009. The difference of optimized surface $CO_2$
flux is calculated as in Fig. 2b. The largest difference of optimized surface $CO_2$ fluxes
between the two experiments occurs in the Conifer Forest ecoregion of Siberia. Compared to
the CNTL experiment, the uptake of optimized surface $CO_2$ flux in Siberia is reduced in JR
for all years except 2003. In 2003, extreme drought condition occurred in the whole northern
mid-latitudes (Knorr et al., 2007) and Europe (Ciais et al., 2005), which resulted in increased
NEE, i.e., reduced uptake of $CO_2$, in EB in the CNTL experiment. The uptake of optimized
surface $CO_2$ fluxes in Siberia in 2003 is reduced in the CNTL experiment due to the remote
effect of drought in Europe. Compared to the CNTL experiment, the uptake of optimized
surface $CO_2$ fluxes in Siberia in 2003 is not reduced that much in the JR experiment due to the
assimilation of the JR-STATION data in Siberia. Despite the number of JR-STATION data
used in the optimization in 2003 being relatively smaller than that in the later experiment
period, new observations in the JR experiment provide information on the uptake of
optimized surface $CO_2$ fluxes in 2003 in Siberia (Fig. 3b).

Optimized surface $CO_2$ fluxes averaged from 2002 to 2009 for each ecoregion in the NH are shown in Table 3. In Siberia (EB), optimized surface $CO_2$ uptake from the JR experiment is smaller (larger) than that of the CNTL experiment in the Conifer Forest and Northern Taiga (in other ecoregions). In the Eurasian Temperate (ET), Europe, North American Boreal (NAB), and North American Temperate (NAT) regions, the optimized surface $CO_2$ uptakes from the JR experiment are larger than those of the CNTL experiment in most ecoregions.

Figure 4 shows the time series of annual and average prior and optimized surface $CO_2$ fluxes over global total, global land, and global ocean. For global total, the magnitude of optimized fluxes is much greater than that of prior fluxes due to the greater uptake of optimized fluxes than that of prior fluxes over global land (Figs. 4a and b). In contrast, the magnitude of optimized fluxes over global ocean is slightly weaker than that of prior fluxes (Fig. 4c). As shown in Table 2, the differences between annual and average optimized surface $CO_2$ fluxes over the globe are small and the average is almost the same for the two experiments (Fig. 4a) with a similar trend of -0.33 Pg C $yr^{-2}$ and -0.35 Pg C $yr^{-2}$ in the CNTL and JR experiment respectively, and the differences in global land and ocean are also small (Figs. 4b, c) with a similar trend of -0.22 Pg C $yr^{-2}$ in global land for both the CNTL and JR experiments and -0.11 Pg C $yr^{-2}$ and -0.13 Pg C $yr^{-2}$ in global ocean for the CNTL and JR experiments, respectively. The optimized surface $CO_2$ fluxes from each experiment show similar interannual variability, which implies that the additional Siberian observations do not affect the interannual variability of global surface $CO_2$ uptake.

Figure 5 is the same as Fig. 4 but covers land regions in the NH. Although the optimized surface $CO_2$ fluxes over global total are similar, those over each TransCom region are different in each experiment. The optimized fluxes over each region show greater annual uptake relative to the prior fluxes in both experiment. As expected, the difference between the two experiments is largest in the EB (Fig. 5a) where the new additional observations are assimilated. The JR experiment exhibits a weaker surface $CO_2$ uptake in the EB than does the CNTL experiment except for 2003 as shown in Fig. 3b, whereas the JR experiment exhibits a greater surface $CO_2$ uptake in the other regions, especially over Europe in 2008 and 2009, than the CNTL experiment (Figs. 5b, c, d, and e). It is driven by the increase of $CO_2$ uptake in Eastern Europe (Figs. 3g and h). Because most of JR-STATION sites are located in the western part of Siberia (Fig. 1), the optimized surface $CO_2$ fluxes over Eastern Europe could be affected by JR-STATION observations. The trend of EB in the CNTL experiment is -0.06

Pg C yr$^{-2}$, whereas that in the JR experiment is 0.02 Pg C yr$^{-2}$ due to the reduced uptake of
$CO_2$ in the JR experiment since 2005 (Fig 5a). As a result, the trends of the surface $CO_2$
uptake of EB and Europe in the two experiments show opposite signs. In contrast, the surface
$CO_2$ uptake trends of other land regions in NH are similar between the two experiments.
Figure 6 shows monthly prior and optimized surface $CO_2$ fluxes averaged from 2002 to 2009
with their uncertainties from both experiments. In general, optimized fluxes in both
experiments show greater uptake in boreal summer and weaker uptake in other seasons
compared to the prior fluxes, which results in greater annual $CO_2$ uptake of optimized fluxes
than prior fluxes as shown in Fig. 5. The largest difference in surface $CO_2$ flux between the
two experiments occurs in June and July, which represent the active season of the terrestrial
ecosystem with a large surface $CO_2$ flux uncertainty. The JR experiment exhibits a weaker
surface $CO_2$ summer uptake in the EB (Fig. 6a) and slightly greater uptake in the other
regions (Figs. 6b, c, d, and e). These additional JR-STATION data provides information on
the surface $CO_2$ uptake by vegetation activities in the NH summer.

## 3.2   Comparison with observations

Table 4 presents the average bias of the model $CO_2$ concentrations calculated by the
background and optimized fluxes of the two experiments at each observation site located in
Asia and Europe from 2002 to 2009. The bias is calculated by subtracting the observed $CO_2$
concentrations from the model $CO_2$ concentrations. Biases of the JR experiment are smaller
than those of the CNTL experiment at the JR-STATION sites, which indicates that the
optimized surface $CO_2$ flux of the JR experiment is more consistent with the observed $CO_2$
concentrations than that in the CNTL experiment. The negative bias at five JR-STATION
sites (DEM, IGR, KRZ, NOY, and YAK shown in Fig. 1 and Table 1) located in the forest
area of the EB is reduced compared with those of the CNTL experiment, which indicates that
the optimized surface $CO_2$ uptake of the CNTL experiment is overestimated with respect to
$CO_2$ concentration observations in Siberia. Otherwise, the reduced surface $CO_2$ uptake of the
JR experiment exhibits more consistent model $CO_2$ concentrations in this region. In addition
to the average bias for the entire period, the time series of monthly averaged bias of the model
$CO_2$ concentrations from the observed $CO_2$ concentrations at JR-STATION sites shows that
the JR experiment consistently shows smaller biases compared to the CNTL experiment (not
shown), which implies that the model representation of $CO_2$ at JR-STATION sites is more
accurate in the JR experiment than in the CNTL experiment. Model $CO_2$ concentrations

calculated by background surface $CO_2$ fluxes in the JR experiment are also more consistent with the observations, implying that background scaling factors of the JR experiment are more accurate than those of the CNTL experiment. The background surface $CO_2$ fluxes are calculated by multiplying the background scaling factor with prior biosphere and ocean fluxes as in Eq. (1). In addition, the average innovation $\chi^2$-statistics at the JR-STATION sites are generally close to 1, implying that the defined MDM is an appropriate value. Therefore, by assimilating JR-STATION observation data, the JR experiments exhibits better results than the CNTL experiment at observation sites in EB.

However, at observation sites in ET and Europe, the difference in biases of the two experiments is relatively small and not significant enough to determine which experiment exhibits better results. This is due to the small difference of optimized surface $CO_2$ fluxes between the two experiments in the ET region. The observation sites in Europe are located far from Eastern Europe and Siberia as shown in Fig. 1 so that they are not sensitive to the change of surface $CO_2$ uptake in those regions. In addition, the MDM at four sites (BAL, BSC, HUN, and OBN) in Europe is assigned as 7.5 ppm, the largest value in CarbonTracker, due to poor representation of the transport model at these sites (Peters et al., 2010).

In addition, model $CO_2$ concentrations calculated by optimized fluxes of the two experiments are compared with independent, not assimilated, vertical profiles of $CO_2$ concentration measurements by aircraft at the BRZ site in Siberia. Aircraft measurements were conducted in the afternoon on good weather days. The frequency of flight was usually two to four times per month (Sasakawa et el., 2013). Table 5 presents the average bias, root-mean-square difference (RMSD), mean absolute error (MAE), and Pearson's correlation coefficient of the model $CO_2$ concentrations calculated by optimized fluxes of the two experiments based on the observations at BRZ site as the reference. The statistics are calculated at each vertical bin at 500 meter intervals by using aircraft measurements observed between 1200 – 1600 LST. Overall, the biases of the two experiments are less than 0.80 ppm showing good consistency between model and observed $CO_2$ concentrations. Near the surface, the result of the JR experiment is better than that of CNTL experiment in terms of bias. The bias of the JR experiment is smaller than those of the CNTL experiment at the level under 500 m, whereas the biases of the CNTL experiment are smaller than those of the JR experiment at the levels above 500 m. More $CO_2$ concentrations are generated over the BRZ site because of the reduced uptake of surface $CO_2$ fluxes over Siberia in the JR experiment. The standard

deviations of the CNTL experiment are greater than those of the JR experiment, which implies that the biases of the CNTL experiment fluctuate about its average more than those of the JR experiment. In contrast, the RMSD and MAE of the JR experiment are smaller than those of the CNTL experiment, and the correlation coefficient of the JR experiment is greater than that of the CNTL experiments. Therefore, overall the statistics show that the model $CO_2$ concentrations of the JR experiment are relatively more consistent with independent $CO_2$ concentration observations compared to those of the CNTL experiment over Siberia.

### 3.3  Uncertainty reduction and observation impact

The effects of additional observations on the optimized surface $CO_2$ flux and associated uncertainties are investigated. Figure 7 shows the average uncertain reduction from 2002 to 2009, average in summer (June, July, and August) and average in winter (December, January, February) uncertainty reductions from 2002 to 2009. The uncertainty reduction based on the uncertainty of CNTL as the reference is calculated as

$$\mathrm{UR} = \frac{\sigma_{CNTL} - \sigma_{JR}}{\sigma_{CNTL}} \times 100 (\%) , \tag{8}$$

where $\sigma_{CNTL}$ and $\sigma_{JR}$ are one-sigma standard deviations of the optimized scaling factor for the CNTL experiment and JR experiment, respectively. The maximum uncertainty reduction is the greatest value in any week in the period 2002 to 2009 in each ecoregion. As expected, the average uncertainty reduction is apparent in the Conifer Forest of EB in which JR stations are mainly located, which has the additional observations (Fig. 7a). The uncertainty reduction in Asia and Europe, especially in the forests of Siberia and Eastern Europe, is greater than for other regions. The spatial pattern of the maximum uncertainty reduction is similar to that of the average values from 2002 to 2009 (not shown). The uncertainty reduction of EB in summer is higher than that in winter (Figs. 7b, c) due to a higher uncertainty associated with larger net fluxes in summer compared to winter (Fig. 6a). For example, the average value of the Conifer Forest of EB is 29.1%, the maximum value is 78.6%, the average value in summer is 36.3% and the average value in winter is 29.7%, respectively. The uncertainty reduction of the CNTL and JR experiments based on the prior uncertainty as the reference ($\sigma_{prior}$ used instead of $\sigma_{CNTL}$ in Eq. (8); $\sigma_{CNTL}$ or $\sigma_{JR}$ used instead of $\sigma_{JR}$ in Eq. (8)) shows similar values in the NH except in the Siberia region (not shown). In addition, the difference between average uncertainty reduction of the CNTL and JR experiments based on the prior

uncertainty as the reference (not shown) is very similar to the average of the uncertainty
reduction in Eq. (8) shown in Fig.7a. Therefore, the uncertainties of the optimized surface
$CO_2$ fluxes are reduced by the additional observations.
To investigate the impact of individual observations on the optimized surface $CO_2$ fluxes, the
self-sensitivities are calculated by the method demonstrated by Kim et al. (2014b). The self-
sensitivity is the diagonal element of the influence matrix which measures the impact of
individual observations in the observation space on the optimized surface $CO_2$ flux. A large
self-sensitivity value implies that the information extracted from observations is large. Figure
8 shows the self-sensitivities of the two experiments averaged from 2002 to 2009. The
average self-sensitivities at the JR-STATION sites are approximately 60% larger than those at
the towers in North America, i.e., continuous site category observations in Fig. 1. The global
average self-sensitivities are 4.83% (CNTL experiment) and 5.08% (JR experiment), and the
cumulative impacts for the 5 weeks assimilation window are 18.79% (CNTL experiment) and
19.33% (JR experiment). The average self-sensitivities of additional observations are higher
than those of other sites, providing more information for estimating surface $CO_2$ fluxes. In
particular, the YAK site located in east Siberia provides greater impacts than other JR-
STATION sites located in 60 ~ 90°E.
The RMSDs between the optimized surface $CO_2$ fluxes and the background fluxes at each
assimilation step in summer are calculated (Fig. 9). The RMSD of the analyzed surface $CO_2$
fluxes constrained by one week of observations from the background fluxes in the JR
experiment is greater than that in CNTL experiment (Figs. 9a, b), implying that surface $CO_2$
fluxes in Siberia are analyzed by JR-STATION data in Siberia directly at the first assimilation
step. This is consistent with the high value of self-sensitivities at JR-STATION sites as shown
in Fig. 8b. Because JR-STATION data are abundant and have large self-sensitivities, these
observations provide significant information on the estimated surface $CO_2$ fluxes over Siberia
in the first cycle.  Kim et al. (2014b) showed that the RMSD in Asia increases after 5 weeks
of optimization, which implies that it takes more than 1 week to affect the surface $CO_2$ fluxes
in Siberia by the transport of the $CO_2$ concentrations observed in remote regions. However, by
assimilating the $CO_2$ concentrations observed at the JR-STATION sites in Siberia, the
observation impact on the optimized surface $CO_2$ fluxes in Siberia increases after 1 week of
optimization (Fig. 9b). In contrast, the RMSD in the Siberia region increases after 5 weeks of
optimization in the CNTL experiment compared to that in the JR experiment (Figs. 9c, d),

which corresponds to the reduced uptake of optimized surface $CO_2$ fluxes in JR experiment as shown in Fig. 2b.

### 3.4 Comparison with other results

A comparison of the optimized surface $CO_2$ flux in this study with other previous studies is presented in Table 6. In the EB, the land sink from the JR experiment (-0.77±0.70 Pg C yr$^{-1}$) is smaller than those reported by Zhang et al. (2014b) (-1.02±0.91 Pg C yr$^{-1}$), Maki et al. (2010) (-1.46±0.41 Pg C yr$^{-1}$), and the CT2013B (CarbonTracker released on 9 February 2015; documented online at http://www.esrl.noaa.gov/gmd/ccgg/carbontracker/CT2013B/) results (-1.00±3.75 Pg C yr$^{-1}$), but higher than those reported by Saeki et al. (2013) (-0.35±0.61 Pg C yr$^{-1}$; including biomass burning 0.11 Pg C yr$^{-1}$)), and similar with those reported by Dolman et al. (2012) (-0.613 Pg C yr$^{-1}$).

Because CT2013B and Zhang et al. (2014b) use an inversion framework similar to that in this study, the reduced land sink is caused by assimilating additional observations. The difference in the land sink between the JR experiment and Saeki et al. (2013) is caused by a different inversion system framework (i.e., prior flux information, atmospheric transport model, observation data set, and inversion method) between two studies. Despite the different inversion system framework used in each study, the two studies using the JR-STATION data exhibit similar results in relative terms, reduced uptake of $CO_2$ fluxes and uncertainties over Siberia. Nevertheless, the land sink from the JR experiment is somewhat different from other inversion results, as its value falls within the flux uncertainty range. Although the land sink in Dolman et al. (2012) is the average land sink obtained from three methods (inventory-based, eddy covariance, and inversion methods) and estimated not only for Siberia but for Russian territory including Ukraine, Belarus, and Kazakhstan, the land sinks of the JR experiment and Dolman et al. (2012) shows similar values. Overall, the optimized surface $CO_2$ fluxes in EB of the JR experiment are comparable to those studies mentioned above.

In Europe, though the long-term average land sink from the JR experiment (-0.37±0.64 Pg C yr$^{-1}$) is higher in magnitude than that of CTE2014 (-0.07±0.49 Pg C yr$^{-1}$), the average land sink from 2008-2009 of the JR experiment (-0.75±0.63 Pg C yr$^{-1}$) is much higher in magnitude than that of CTE2014 (-0.11±0.38 Pg C yr$^{-1}$). The land sinks of the JR experiment in 2008 and 2009 are -0.73±0.41 and -0.76±0.38 Pg C yr$^{-1}$, respectively, whereas much lower uptakes (-0.21±0.49, -0.38±0.44 Pg C yr$^{-1}$) are obtained for the CNTL experiment. According

to Reuter et al. (2014), despite the different experiment period, the land sink of Europe in
2010 (-1.02±0.30 Pg C yr$^{-1}$) estimated by using satellite observations is much higher than
previous inversion studies (e.g., Peylin et al. 2013) using only surface observations.

## 4    Summary and conclusions

In this study, to investigate the effect of the Siberian observations, which were not used in the
previous studies using CarbonTracker, on the optimization of surface $CO_2$ fluxes, two
experiments, named CNTL and JR, with different sets of observations from 2000 to 2009
were conducted and optimized surface $CO_2$ fluxes from 2002 to 2009 were analyzed.
The global balances of the sources and sinks of surface $CO_2$ fluxes were maintained with a
similar trend for both experiments, while the distribution of the optimized surface $CO_2$ fluxes
changed. The magnitude of the optimized biosphere surface $CO_2$ uptake and its uncertainty in
EB (Siberia) was decreased from -1.17±0.93 Pg C yr$^{-1}$ to -0.77±0.70 Pg C yr$^{-1}$, whereas it was
increased in other regions of the NH (Eurasian Temperate, Europe, North American Boreal,
and North American Temperate). The land sink of Europe for 2008 and 2009 of the JR
experiment increased significantly from -0.30±0.68 Pg C yr$^{-1}$ to -0.75±0.63 Pg C yr$^{-1}$, which
is consistent with the other inversion results (Reuter et al., 2014) inferred by satellite
observations. Additional observations are used to correct the surface $CO_2$ uptake in June and
July, the active vegetation uptake season, in terms of monthly average optimized surface $CO_2$
fluxes. As a result, the additional observations do not exhibit a change in the magnitude of the
global surface $CO_2$ flux balance because they provide detailed information about the Siberian
land sink instead of the global land sink magnitude, when they are used in our inversion
modeling system (i.e., CarbonTracker).
The model $CO_2$ concentrations using the background and optimized surface $CO_2$ fluxes in the
JR experiment are more consistent with the $CO_2$ observations used in the optimization than
those in the CNTL experiment, showing lower biases in the EB region. In contrast, the
differences of biases of the two experiments in ET and Europe are smaller than those in EB.
In comparison with vertical profiles of $CO_2$ concentration observations which are not used in
the optimization, the model $CO_2$ concentrations in the JR experiment show smaller RMSD
and MAE values, and a higher correlation coefficient that those in CNTL experiment.

The new observations provide information on the optimized surface $CO_2$ fluxes. The observation impact of the Siberian observation data is investigated by means of uncertainty reduction and self-sensitivity calculated by an influence matrix. Additional observations reduce the uncertainty of the optimized surface $CO_2$ fluxes in Asia and Europe, mainly in the EB region (Siberia), where the new observations are used in the assimilation. The average self-sensitivities of the JR-STATION sites are approximately 60% larger than those for other continuous measurements (e.g., tower measurements in North America). The global average self-sensitivity and cumulative impact of the JR experiment are higher than those of the CNTL experiment, which implies that the impact of JR-STATION data on optimized surface $CO_2$ fluxes is higher than that of other observations used in both the CNTL and JR experiments. The RMSD of the analyzed surface $CO_2$ fluxes constrained by one week of observations from the background fluxes also suggests that new Siberian observations provide information on the optimized surface $CO_2$ fluxes.

This study shows that the JR-STATION data affect the longitudinal distribution of the total NH sinks, especially in the EB and Europe, when it is used by atmospheric $CO_2$ inversion modeling. In the future, it is expected that Siberian observations will be used as an important constraint for estimating surface $CO_2$ fluxes over the NH with various $CO_2$ observations (e.g., satellite and aircraft measurements) simultaneously.

**Acknowledgements**

The authors appreciate Dr. William Lahoz, Dr. Abhishek Chatterjee, and other reviewers for their valuable comments. This study was funded by the Korea Meteorological Administration Research and Development Program under the Grant KMIPA 2015-2021. The JR-STATION is supported by the Global Environment Research Account for National Institutes of the Ministry of the Environment, Japan and the Russian Foundation for Basic Research (Grant No. 14-05-00590). The authors also acknowledge atmospheric $CO_2$ measurements data providers and cooperating agencies at China Meteorological Administration, Commonwealth Scientific and Industrial Research Organization, Environment Canada, Finnish Meteorological Institute, Hungarian Meteorological Service, Japan Meteorological Agency, Lawrence Berkeley National Laboratory, National Institute of Environmental Research, Norwegian Meteorological Institute, Max Planck Institute for Biogeochemistry, Morski Instytut Rybacki, National Center for Atmospheric Research, National Oceanic and Atmospheric Administration Earth System Research Laboratory, and Romanian Marine Research Institute.

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

Table 1. Information on observation sites located in the Asia and Europe region. MDM
represents the model-data mismatch which is the observation error.

| Site | Location | Latitude | Longitude | Height (Sampling height) (m) | Laboratory (Cooperating agency) | MDM (ppm) |
|---|---|---|---|---|---|---|
| AZV | Azovo, Russia | 54.71°N | 73.03°E | 110(50) | NIES | 3 |
| BRZ | Berezorechka, Russia | 56.15°N | 84.33°E | 168(80) | NIES | 3 |
| DEM | Demyanskoe, Russia | 59.79°N | 70.87°E | 63(63) | NIES | 3 |
| IGR | Igrim, Russia | 63.19°N | 64.41°E | 9(47) | NIES | 3 |
| KRS | Karasevoe, Russia | 58.25°N | 82.42°E | 76(67) | NIES | 3 |
| NOY | Noyabrsk, Russia | 63.43°N | 75.78°E | 108(43) | NIES | 3 |
| SVV | Savvushka, Russia | 51.33°N | 82.13°E | 495(52) | NIES | 3 |
| VGN | Vaganovo, Russia | 54.50°N | 62.32°E | 192(85) | NIES | 3 |
| YAK | Yakutsk, Russia | 62.09°N | 129.36°E | 264(77) | NIES | 3 |
| WLG | Mt. Waliguan, China | 36.29°N | 100.9°E | 3810 | CMA/ESRL | 1.5 |
| BKT | Bukit Kototabang, Indonesia | 0.20°S | 100.32°E | 864 | ESRL | 7.5 |
| WIS | Sede Boker, Israel, | 31.13°N | 34.88°E | 400 | ESRL | 2.5 |
| KZD | Sary Taukum, Kazakhstan | 44.45°N | 77.57°E | 412 | ESRL | 2.5 |
| KZM | Plateau Assy, Kazakhstan | 43.25°N | 77.88°E | 2519 | ESRL | 2.5 |
| TAP | Tae-ahn Peninsula, South Korea | 36.73°N | 126.13°E | 20 | ESRL | 5 |
| UUM | Ulaan Uul, Mongolia | 44.45°N | 111.10°E | 914 | ESRL | 2.5 |
| CRI | Cape Rama, India | 15.08°N | 73.83°E | 60 | CSIRO | 3 |
| LLN | Lulin, Taiwan | 23.47°N | 120.87°E | 2862 | ESRL | 7.5 |
| SDZ | Shangdianzi, China | 40.39°N | 117.07°E | 287 | CMA/ESRL | 3 |
| MNM | Minamitorishima, Japan | 24.29°N | 153.98°E | 8 | JMA | 3 |
| RYO | Ryori, Japan | 39.03°N | 141.82°E | 260 | JMA | 3 |
| YON | Yonagunijima, Japan | 24.47°N | 123.02°E | 30 | JMA | 3 |
| GSN | Gosan, South Korea | 33.15°N | 126.12°E | 72 | NIER | 3 |
| BAL | Baltic Sea, Poland | 55.35°N | 17.22°E | 3 | ESRL (MIR[*]) | 7.5 |
| BSC | Black Sea, Constanta, Romania | 44.17°N | 28.68°E | 3 | ESRL (RMRI[*]) | 7.5 |
| HUN | Hegyhatsal, Hungary | 46.95°N | 16.65°E | 248 | ESRL (HMS[*]) | 7.5 |
| OBN | Obninsk, Russia | 55.11°N | 36.60°E | 183 | ESRL | 7.5 |
| OXK | Ochsenkopf, Germany | 50.03°N | 11.80°E | 1022 | ESRL (MPI-BGC[*]) | 2,5 |
| PAL | Pallas-Sammaltunturi, GaW Station, Finland | 67.97°N | 24.12°E | 560 | ESRL (FMI[*]) | 2.5 |
| STM | Ocean Station M, Norway | 66.00°N | 2.00°E | 0 | ESRL (MET Norway[*]) | 1.5 |

*Cooperating agencies of observation sites in Europe: Morski Instytut Rybacki (MIR), Romanian Marine
Research Institute (RMRI), Hungarian Meteorological Service (HMS), Max Planck Institute for
Biogeochemistry (MPI-BGC), Finnish Meteorological Institute (FMI), Norwegian Meteorological Institute
(MET Norway).
Table 2. A prior and optimized surface $CO_2$ fluxes and their one-sigma uncertainties (Pg C
yr$^{-1}$ Region$^{-1}$) of global total, land, ocean, and other regions averaged spatially from 2002 to
2009. The CNTL experiment is conducted by using set of observations without
observations in the Siberia region, whereas the JR experiment is conducted using all
available observations including the Siberia data.

| Region | A priori | CNTL | JR. |
|---|---|---|---|
| Eurasian Boreal | -0.07±1.10 | -1.17±0.93 | -0.77±0.70 |
| Eurasian Temperate | -0.05±0.49 | -0.31±0.41 | -0.36±0.40 |
| Europe | -0.01±-0.76 | -0.20±0.67 | -0.37±0.64 |
| North American Boreal | -0.04±0.61 | -0.30±0.38 | -0.36±0.38 |
| North American Temperate | -0.02±0.66 | -0.55±0.41 | -0.59±0.41 |
| Northern Hemisphere total | -1.42±1.85 | -3.21±1.49 | -3.21±1.34 |
| Tropical total | 0.06±0.80 | 0.12±0.74 | 0.11±0.74 |
| Southern Hemisphere total | -2.57±0.97 | -2.46±0.81 | -2.45±0.81 |
| Global total | -3.94±2.24 | -5.54±1.85 | -5.55±1.72 |
| Global land | -1.33±1.90 | -3.59±1.57 | -3.52±1.43 |
| Global ocean | -2.61±1.19 | -1.95±0.97 | -2.03±0.96 |

Table 3. The optimized surface $CO_2$ fluxes (Pg C $yr^{-1}$ $Region^{-1}$) of ecosystem types at Eurasian Boreal, Eurasian Temperate, Europe, North
American Boreal, and North American Temperate region averaged over 2002 - 2009. The CNTL experiment is conducted by using set of
observations without observations in the Siberia region, whereas the JR experiment is conducted using all available observations including
the Siberia data.

| Ecosystem type | Eurasian Boreal | | Eurasian Temperate | | Europe | | North American Boreal | | North American Temperate | |
|---|---|---|---|---|---|---|---|---|---|---|
| | CNTL | JR | CNTL | JR | CNTL | JR | CNTL | JR | CNTL | JR |
| Conifer Forest | -0.815 | -0.337 | -0.005 | -0.005 | -0.067 | -0.069 | -0.107 | -0.121 | -0.054 | -0.069 |
| Broadleaf Forest | -0.006 | -0.013 | -0.004 | -0.005 | -0.005 | -0.005 | 0.000 | 0.000 | -0.002 | -0.002 |
| Mixed Forest | -0.049 | -0.090 | -0.029 | -0.034 | -0.025 | -0.063 | -0.053 | -0.054 | -0.019 | -0.021 |
| Grass/Shrub | -0.035 | -0.056 | -0.247 | -0.285 | -0.016 | -0.032 | 0.000 | -0.001 | -0.077 | -0.081 |
| Tropical Forest | 0.000 | 0.000 | -0.001 | -0.001 | 0.000 | 0.000 | 0.000 | 0.000 | 0.000 | 0.000 |
| Scrub/Woods | 0.000 | 0.000 | -0.002 | -0.002 | -0.001 | -0.001 | 0.000 | 0.000 | -0.013 | -0.013 |
| Semitundra | -0.145 | -0.188 | -0.007 | -0.009 | -0.008 | -0.009 | -0.057 | -0.086 | -0.010 | -0.011 |
| Fields/Woods/Savanna | -0.012 | -0.021 | -0.005 | -0.005 | 0.003 | -0.009 | -0.004 | -0.004 | -0.149 | -0.153 |
| Northern Taiga | -0.094 | -0.029 | 0.000 | 0.000 | -0.006 | -0.007 | -0.066 | -0.077 | 0.000 | 0.000 |
| Forest/Field | -0.003 | -0.008 | 0.006 | 0.006 | -0.086 | -0.105 | -0.001 | -0.001 | -0.012 | -0.016 |
| Wetland | -0.002 | -0.014 | 0.000 | -0.000 | -0.001 | -0.002 | -0.003 | -0.006 | -0.002 | -0.003 |
| Shrub/Tree/Suc | 0.000 | 0.000 | -0.001 | -0.001 | 0.000 | 0.000 | 0.000 | 0.000 | 0.000 | 0.000 |
| Crops | -0.002 | -0.008 | -0.019 | -0.022 | -0.007 | -0.075 | 0.000 | 0.000 | -0.216 | -0.227 |
| Wooded tundra | -0.003 | -0.005 | 0.000 | 0.000 | 0.003 | 0.003 | -0.003 | -0.002 | 0.000 | 0.000 |
| Water | 0.000 | 0.000 | 0.000 | 0.000 | 0.000 | 0.000 | -0.001 | -0.001 | -0.001 | -0.001 |

Table 4. Average differences between model $CO_2$ concentrations (ppm) simulated using the background and the observed $CO_2$ concentration (ppm) (fourth and sixth columns), model $CO_2$ concentrations (ppm) simulated using the optimized surface $CO_2$ flux and the observed $CO_2$ concentration (ppm) (fifth and seventh columns), and average innovation $\chi^2$ from 2002 to 2009 at observation sites located in Asia and Europe (eighth column). The CNTL experiment is conducted by using set of observations without observations in the Siberia region, whereas the JR experiment is conducted using all available observations including the Siberia data.

| Region | Site | MDM [ppm] | CNTL | | JR | | Innovation $\chi^2$ |
| --- | --- | --- | --- | --- | --- | --- | --- |
| | | | Bias (background) | Bias (optimized) | Bias (background) | Bias (optimized) | |
| Eurasian Boreal | AZV | 3 | 1.68 | 1.04 | 0.77 | 0.19 | 0.85 |
| | BRZ | 3 | 1.41 | 0.68 | 0.67 | 0.39 | 1.17 |
| | DEM | 3 | 0.15 | -0.84 | 0.32 | 0.11 | 0.84 |
| | IGR | 3 | -1.58 | -2.71 | -0.52 | -1.26 | 1.15 |
| | KRS | 3 | 0.57 | -0.22 | 0.27 | 0.12 | 1.22 |
| | NOY | 3 | -0.02 | -1.06 | 0.16 | 0.00 | 0.86 |
| | SVV | 3 | 1.25 | 0.71 | 0.63 | 0.09 | 0.96 |
| | VGN | 3 | 2.55 | 2.11 | 1.50 | 0.84 | 1.18 |
| | YAK | 3 | 0.23 | -2.18 | 0.87 | 0.03 | 1.36 |
| Eurasian Temperate | WLG | 1.5 | 0.17 | 0.19 | 0.15 | 0.16 | 1.09 |
| | BKT | 7.5 | 4.12 | 4.06 | 4.13 | 4.05 | 0.57 |
| | WIS | 2.5 | 0.27 | 0.12 | 0.22 | 0.07 | 0.72 |
| | KZD | 2.5 | 1.79 | 0.98 | 1.42 | 1.14 | 1.26 |
| | KZM | 2.5 | 1.17 | 0.96 | 1.13 | 0.93 | 1.26 |
| | TAP | 5 | 0.50 | 0.55 | 0.58 | 0.71 | 0.58 |
| | UUM | 2.5 | 0.24 | -0.07 | 0.20 | 0.12 | 1.05 |
| | CRI | 3 | -1.95 | -1.57 | -1.94 | -1.56 | 0.66 |
| | LLN | 7.5 | 4.42 | 3.09 | 4.42 | 3.09 | 0.47 |
| | SDZ | 3 | -3.02 | -5.26 | -3.09 | -5.28 | 2.08 |
| | MNM | 3 | 0.56 | 0.52 | 0.59 | 0.56 | 0.17 |
| | RYO | 3 | 1.26 | 1.16 | 1.32 | 1.32 | 1.07 |
| | YON | 3 | 1.10 | 0.98 | 1.14 | 1.07 | 0.56 |
| | GSN | 3 | -1.92 | -1.71 | -1.92 | -1.70 | 1.83 |
| Europe | BAL | 7.5 | -1.23 | -1.32 | -1.31 | -1.45 | 0.37 |
| | BSC | 7.5 | -4.12 | -4.97 | -4.12 | -5.13 | 1.01 |
| | HUN | 7.5 | 0.93 | 0.53 | 0.86 | 0.36 | 0.46 |
| | OBN | 7.5 | 0.70 | -0.71 | 0.59 | -0.89 | 0.44 |
| | OXK | 2.5 | 0.50 | 0.02 | 0.43 | -0.09 | 1.52 |
| | PAL | 2.5 | 0.47 | 0.07 | 0.58 | 0.16 | 0.76 |
| | STM | 1.5 | 0.54 | 0.42 | 0.55 | 0.42 | 0.76 |

1    Table 5. Bias, root mean square difference, mean absolute error, and Pearson's Correlation

2    Coefficient of the model $CO_2$ concentration of CNTL and JR experiments in comparison with

3    the vertical profile of $CO_2$ concentrations at BRZ site.

| Altitude (km) | Bias (ppm) | | Root-Mean-Square Difference (ppm) | | Mean Absolute Error (ppm) | | Pearson's Correlation Coefficient | |
|---|---|---|---|---|---|---|---|---|
| | CNTL | JR | CNTL | JR | CNTL | JR | CNTL | JR |
| ~ 0.5 | -0.38±4.73 | -0.05±4.39 | 4.06 | 3.75 | 3.42 | 3.07 | 0.94 | 0.95 |
| 0.5 ~ 1.0 | 0.23±4.05 | 0.42±3.75 | 3.58 | 3.33 | 2.94 | 2.72 | 0.94 | 0.95 |
| 1.0 ~ 1.5 | 0.19±3.80 | 0.31±3.53 | 3.35 | 3.11 | 2.70 | 2.49 | 0.94 | 0.95 |
| 1.5 ~ 2.0 | 0.22±3.38 | 0.33±3.19 | 2.94 | 2.79 | 2.33 | 2.19 | 0.93 | 0.94 |
| 2.0 ~ 2.5 | 0.02±3.19 | 0.08±3.07 | 2.64 | 2.54 | 2.19 | 2.11 | 0.93 | 0.94 |
| 2.5 ~ 3.0 | 0.79±2.84 | 0.80±2.53 | 1.44 | 1.30 | 2.21 | 1.99 | 0.92 | 0.94 |
| 3.0 ~ | 0.61±3.15 | 0.61±2.91 | 1.49 | 1.38 | 2.42 | 2.26 | 0.89 | 0.91 |

Table 6. Optimized surface $CO_2$ fluxes (Pg C $yr^{-1}$) from this study and other inversion studies.

| Citation | Area | Estimate surface $CO_2$ flux | Period | Remarks |
|---|---|---|---|---|
| This study | Eurasian Boreal | -0.77±0.70 | 2002-2009 | JR experiment |
| Saeki et al. (2013) | Eurasian Boreal | -0.35±0.61 | 2000-2009 | Including biomass burning (0.11Pg C $yr^{-1}$), Using JR-STATION observations |
| Zhang et al. (2014b) | Eurasian Boreal | -1.02±0.91 | 2006-2010 | Using CONTRAL observations |
| Maki et al. (2010) | Eurasian Boreal | -1.46±0.41 | 2001-2007 | |
| Dolman et al. (2012) | Russia[a] | -0.613 | | Average of inventory-based, eddy covariance, and inversion methods |
| CT2013B[b] | Eurasian Boreal | -1.00±3.75 | 2002-2009 | |
| This study | Europe | -0.38±0.64 -0.75±0.63 | 2002-2009 2008-2009 | JR experiment |
| Reuter et al. (2014) | Europe | -1.02±0.30 | 2010 | Using satellite data |
| CTE2014[c] | Europe | -0.07±0.49 -0.11±0.38 | 2002-2009 2008-2009 | |

[a]Including Ukraine, Belarus and Kazakhstan (total area is $17.1 \times 10^{12}$ $m^2$)
[b]The results of CT2013B (http://www.esrl.noaa.gov/gmd/ccgg/carbontracker/CT2013B/) were
derived from (ftp://aftp.cmdl.noaa.gov/products/carbontracker/co2/fluxes/).
[c]The results of CTE2014 (CarbonTracker Europe, Peters et al., 2010) were derived from
(ftp://ftp.wur.nl/carbontracker/data/fluxes/).

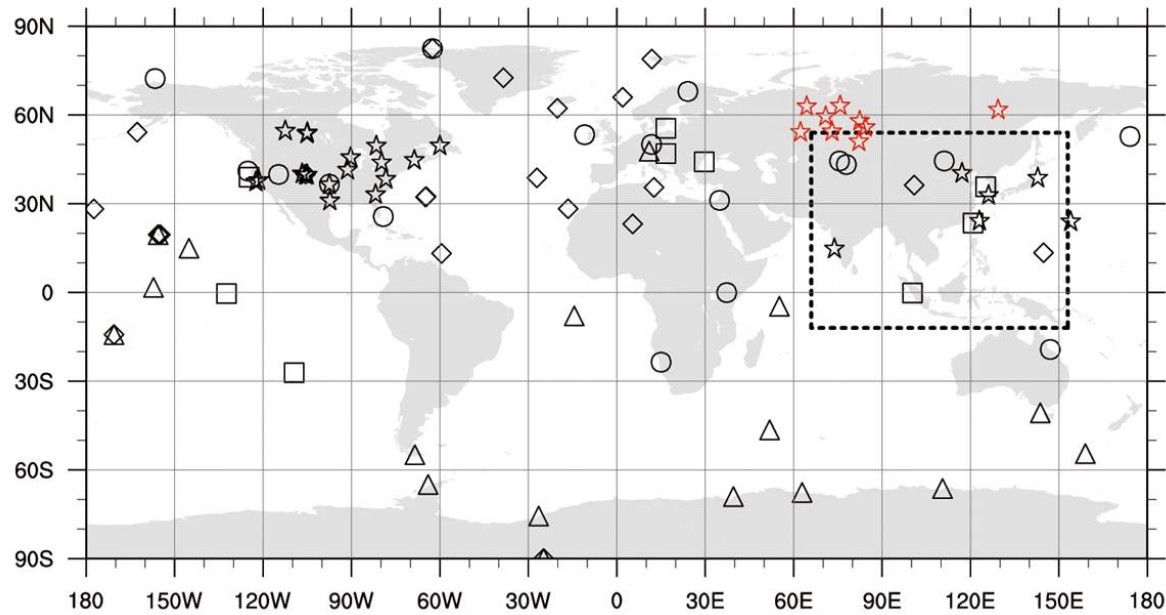

Figure 1. Observation networks of $CO_2$ concentrations around the globe and the nested domain of the TM5 transport model over Asia (dashed box). Each observation site is assigned to different categories (△: MBL; ○: Continental; ◇: Mixed land/ocean and mountain; ☆: Continuous; □: Difficult). JR-STATION observation sites are represented in red color.

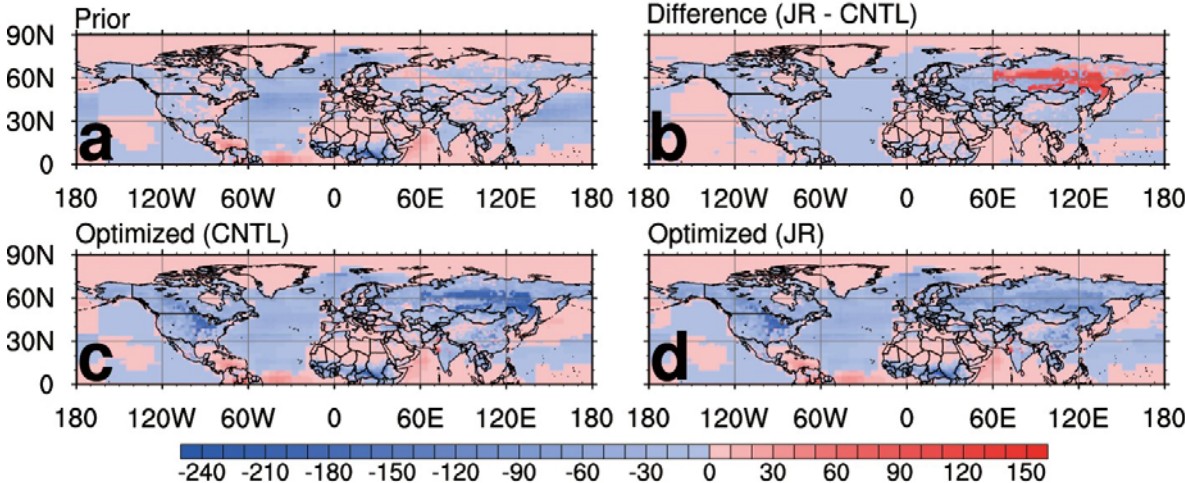

Figure 2. Average biosphere and ocean fluxes (gC m$^{-2}$ yr$^{-1}$) from 2002 to 2009 of (a) the prior
flux, (b) the difference between the optimized fluxes in the JR and CNTL experiments, (c) the
optimized flux in the CNTL experiment, and (d) the optimized flux in the JR experiment.
Blue colors (negative) denote net $CO_2$ flux uptake while red colors (positive) denote net $CO_2$
release to the atmosphere. The difference is calculated by subtracting surface $CO_2$ flux of
CNTL experiment from that of JR experiment.

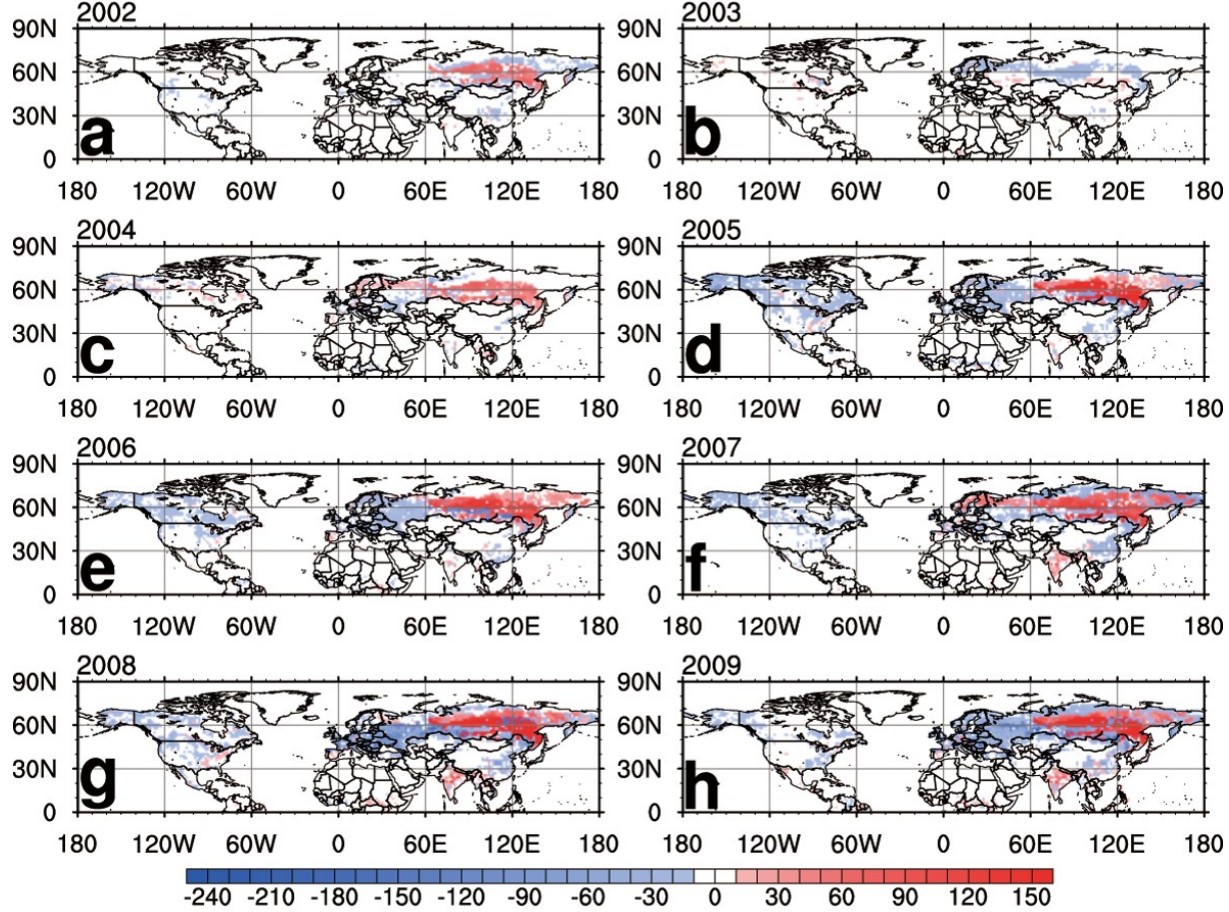

Figure 3. The difference between the optimized biosphere fluxes from the JR and CNTL experiment (g C m$^{-2}$ yr$^{-1}$) of (a) 2002, (b) 2003, (c) 2004, (d) 2005, (e) 2006, (f) 2007, (g) 2008, and (h) 2009. Blue colors (negative) denote net $CO_2$ flux uptake while red colors (positive) denote net $CO_2$ release to the atmosphere. The difference is calculated by subtracting surface $CO_2$ flux of CNTL experiment from that of JR experiment.

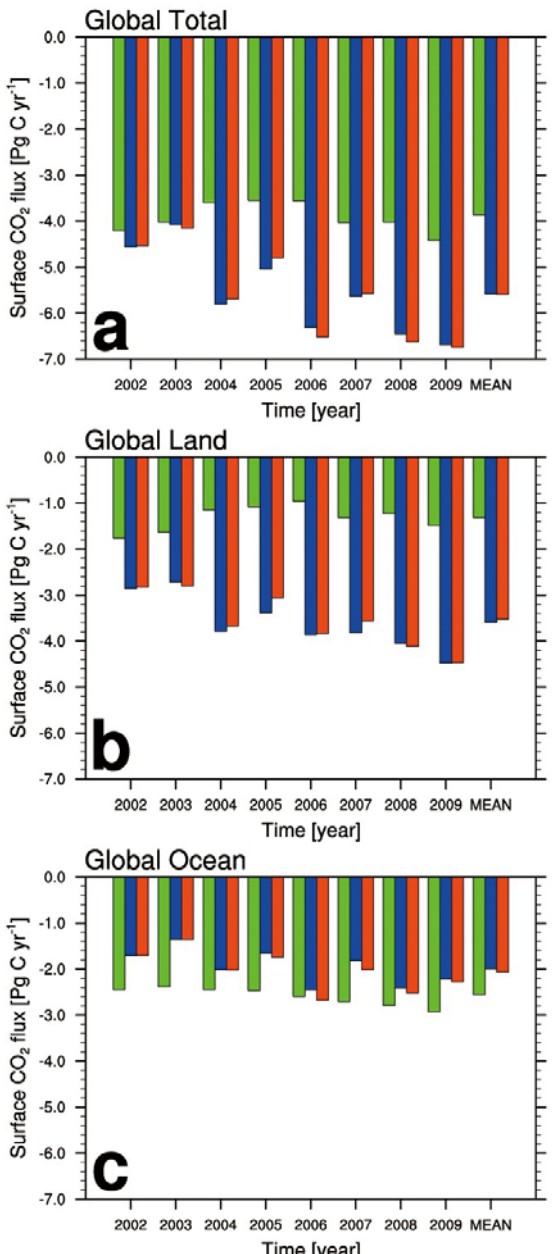

Figure 4. Annual and average biosphere and ocean fluxes (Pg C yr$^{-1}$) from the prior (green bar), CNTL (blue bar) and JR (red bar) experiment aggregated over the (a) whole globe, (b) land, and (c) ocean.

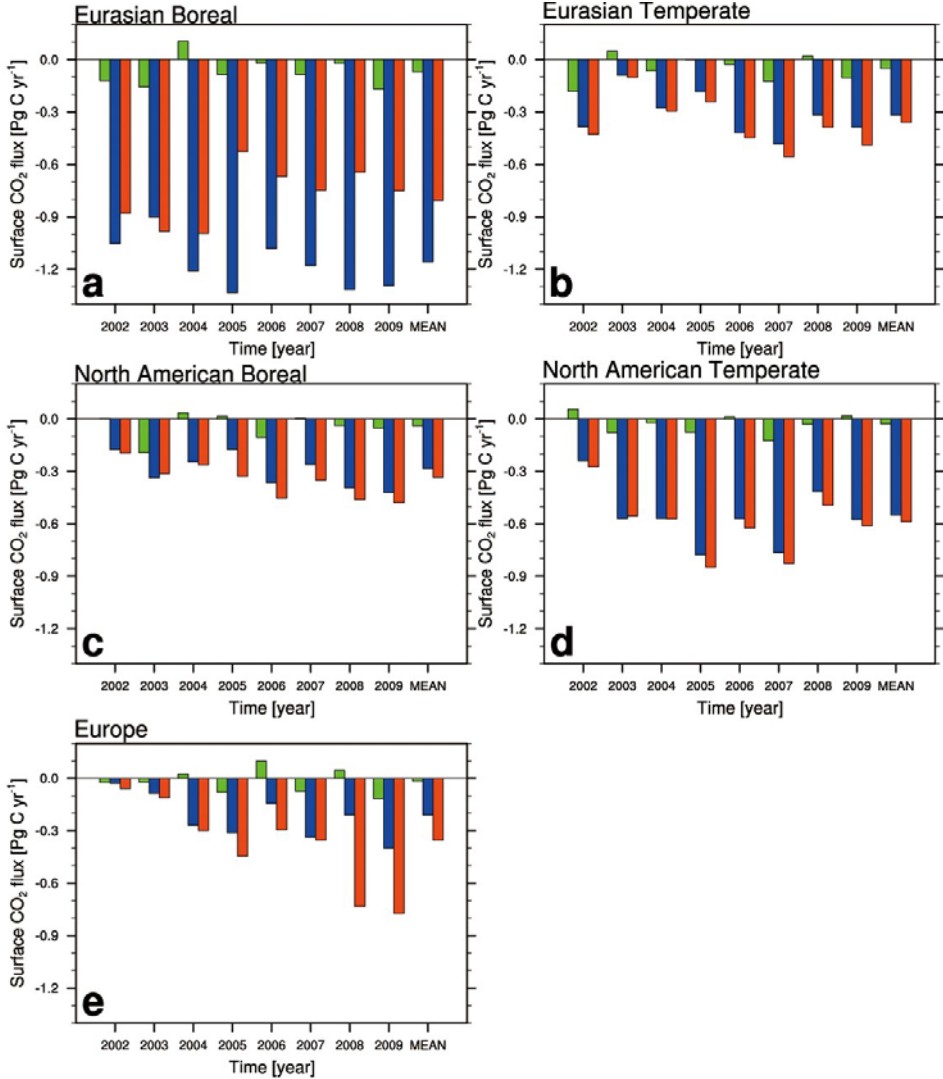

Figure 5. Annual and average biosphere fluxes (Pg C yr$^{-1}$) from the prior (green bar), CNTL (blue bar) and JR (red bar) experiment aggregated over the (a) Eurasian Boreal, (b) Eurasian Temperate, (c) North American Boreal, (d) North American Temperate, and (e) Europe.

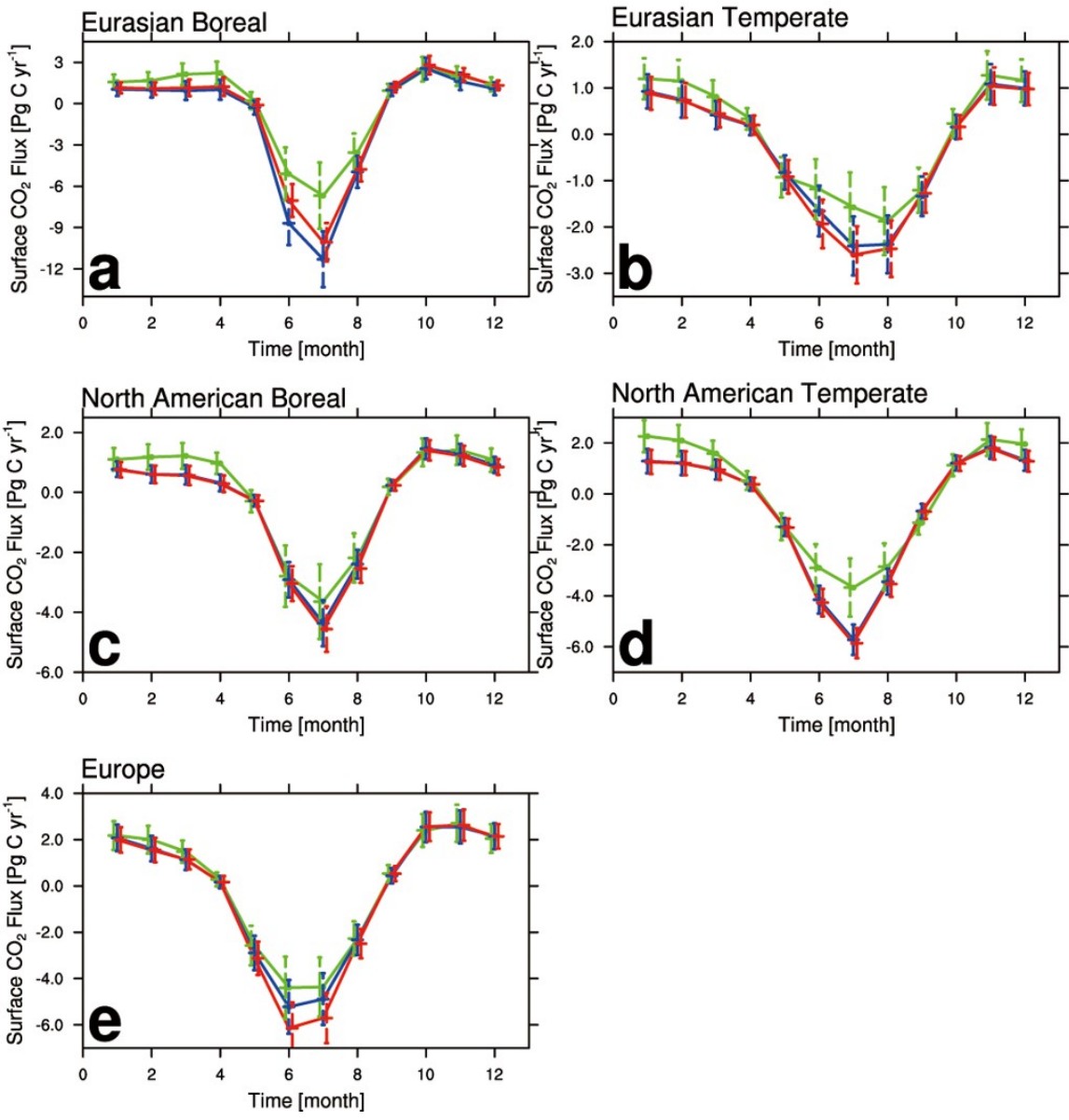

Figure 6. The monthly prior (green) and optimized biosphere fluxes averaged from 2002 to 2009 of CNTL (blue) and JR (red) experiment with their uncertainties over the (a) Eurasian Boreal, (b) Eurasian Temperate, (c) North American Boreal, (d) North American Temperate, and (e) Europe.

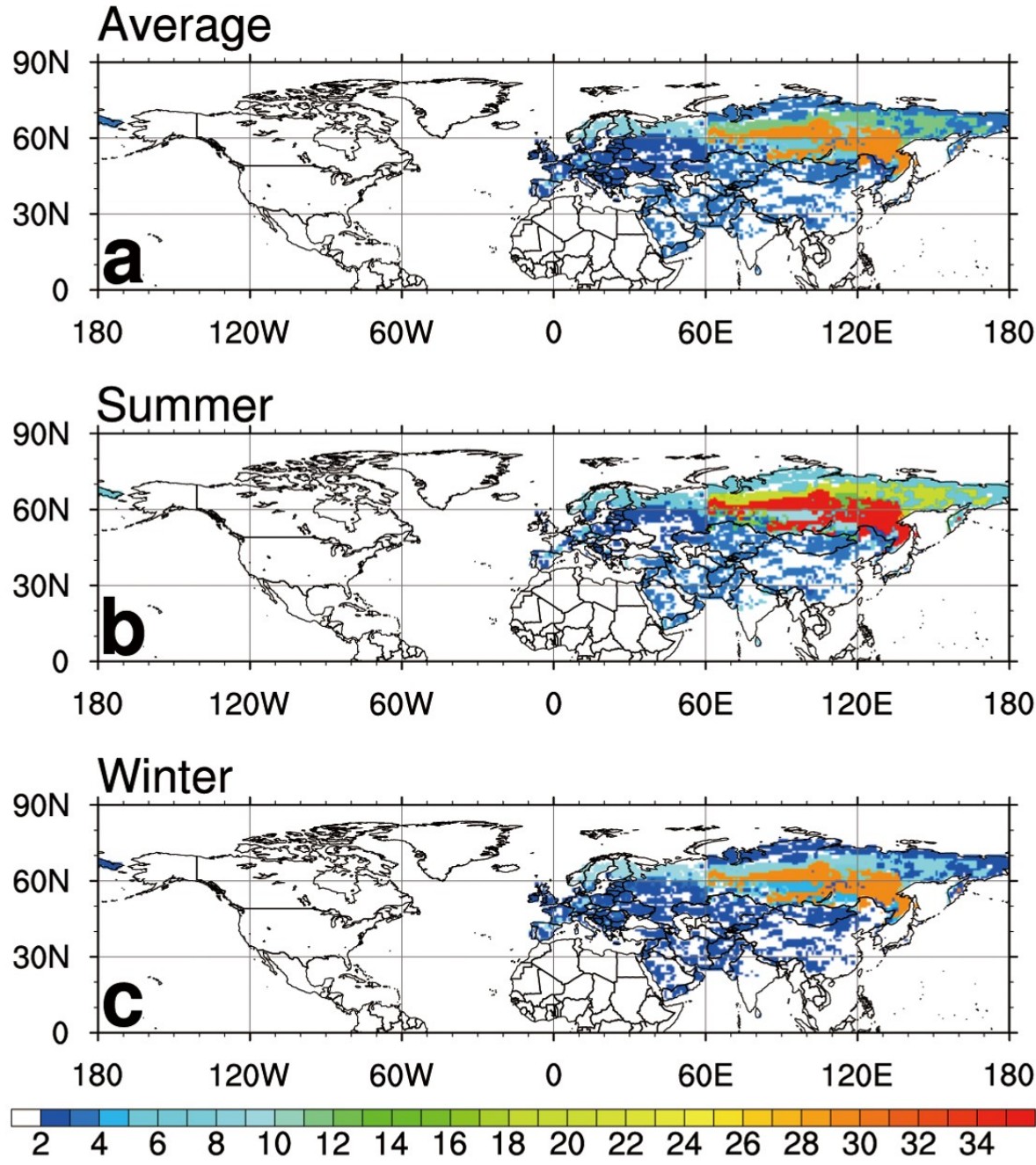

Figure 7. (a) Average uncertainty reduction (%) from 2002 to 2009, average uncertainty reduction (%) in (b) summer, and (c) winter for the estimated uncertainty of the JR experiment relative to that of the CNTL experiment. Red (blue) denotes relatively high (low) values of uncertainty reduction.

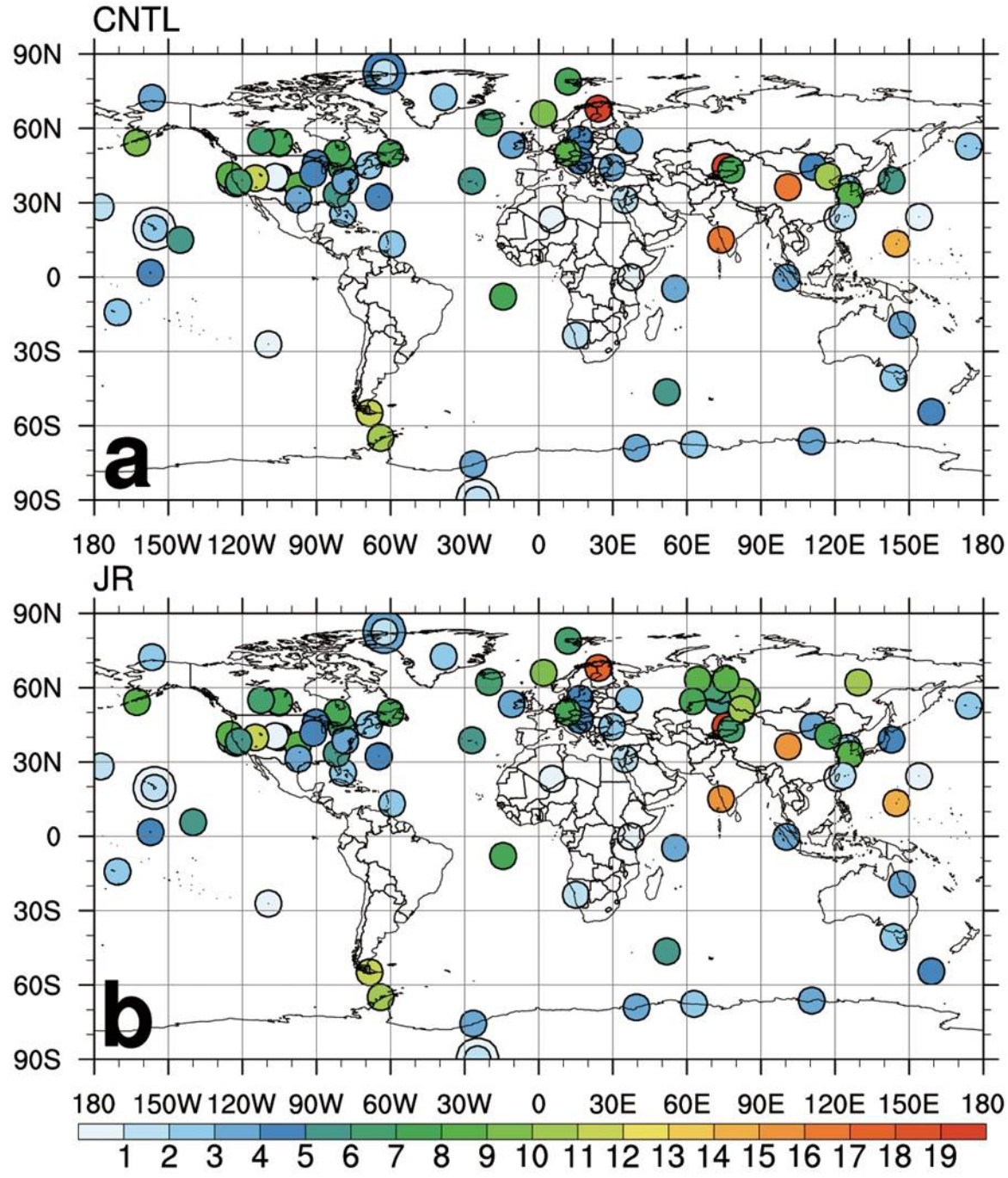

Figure 8. Self-sensitivity at each observation site averaged from 2002 to 2009 of (a) CNTL experiment and (b) JR experiment. The overlapping observation sites at the same locations or at close locations are distinguished by different sizes of circles. Red (blue) denotes relatively high (low) values of self-sensitivity.

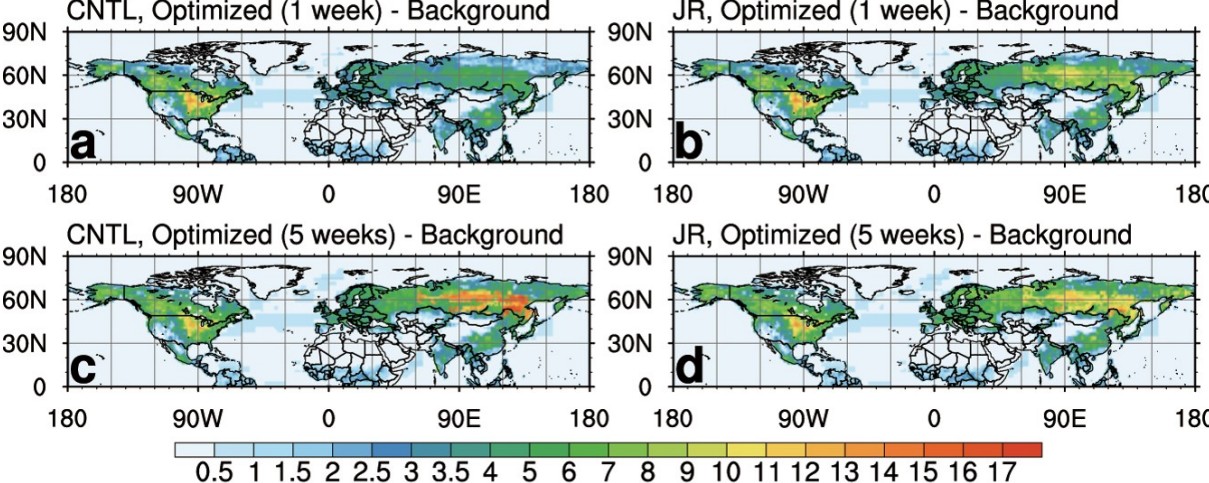

Figure 9. RMSD averaged from 2002 to 2009 between the background flux and posterior flux
optimized in Northern Hemisphere summer by 1 week of observations of (a) CNTL and (b)
JR experiment; and by 5 weeks of observations of (c) CNTL and (d) JR experiment. The units
are g C m$^{-2}$ week$^{-1}$. Red (blue) denotes relatively high (low) value of RMSD.