# Peer review of "Impact of Siberian observations on the optimization of"

_Atmospheric Chemistry and Physics, 2015_

## Referee Comment (RC1) · Anonymous Referee #2 · 22 May 2016

This study evaluates the influence of additional CO2 observations (from the JR-STATION towers) on the analysis of Eurasian and global CO2 surface fluxes. The novelty of this study is in using these additional tower observations, which have not been used within an inverse modeling study before. The results demonstrate that these observations do have a certain amount of impact, namely it adjusts the flux patterns and magnitudes between Eurasian Boreal (local) and other NH land regions (non-local). This is expected based on the way an inverse modeling system works, especially the resultant interplay between the observation density/network and the prior weighed by their respective covariances. This is not a novel finding in itself. What is of greater interest, are the adjustments that are made to the surface fluxes and whether they are correct or not (especially the reduced sink in Eurasian Boreal region). No independent evaluation, either of the posterior CO2 concentrations or the posterior fluxes with any

kind of independent data has been provided, however. The authors have compared their flux estimates to a suite of previous studies. But these studies cover different temporal extent (i.e., span across a wide variety of years), and second all of the estimates fall within the reported uncertainty bounds. There is no rationale behind the authors claim that the flux estimates from the JR experiment are more comparable to the previous studies than the CNTL experiment (Page 14, Lines 4-5). It is also highly misleading that in Section 3.2 the authors show results comparing the posterior CO2 concentrations to the observations that are being assimilated in the first place. Finally in Section 3.3, the uncertainty reduction should be calculated individually for the CNTL and the JR experiments relative to the prior uncertainties that were specified. It is again misleading to compare two posterior uncertainties (without knowing which one provides a baseline) and call this calculation as an "uncertainty reduction".

The following points provide a checklist on critical sensitivity tests/issues that should be addressed to first validate the results presented in this study, and thereby make it relevant and appealing to the carbon science community.
* * *
1) Evaluation of posterior CO2 concentrations with independent data – This is the most important step that is missing from this study. This should be done either by comparison with independent data or via data denial experiments. In the latter case, specific set of in situ observations that are common to both CNTL and JR experiments may be held in reserve (i.e., those data should not be assimilated into the CT system). The posterior CO2 concentrations from the two experiments should be compared to this independent data both qualitatively and quantitatively.

2) Uncertainty estimates associated with the analyzed flux estimates – On Page 9, Lines 13-14, the authors claim that the ". . .uncertainty is calculated as one-sigma standard deviation of the fluxes estimated, using Gaussian errors". It is unclear why the authors choose this approach when they are using an ensemble Kalman filter based

system, where they should be able to directly recover the posterior uncertainty over the entire time period. Why is such an ad-hoc approach used to calculate the uncertainty? What is the basis for this approach?

3) Reduced uptake estimated in EB between 2002-2009 – Possibly the real significant finding from the additional JR-STATION tower observations are that the overall magnitude of the uptake reduced in Eurasian boreal region during NH summer. This is a reasonable conclusion for the summer of 2003 (anomalous drought for this year) but the authors claim a consistent reduction averaged out over the entire 2002-2009 period. The authors do not address any underlying mechanism for this difference in uptake from the two experiments. Is this simply an artifact of the inverse modeling setup, interplay between data density, error covariances, etc.? Or are there changes in vegetative activity that took place during this period in the Eurasian Boreal region and the JR-STATION tower observations were able to observe those local changes. The authors need to provide some form of mechanistic understanding for their inverse modeling results.

4) Prior flux estimates and associated uncertainty used throughout the study – For Figures 4,5 and 6 the authors should add the prior flux estimates (say green or gray bars/lines) to the figures. For the uncertainty reduction reported in Section 3.3, the authors should use the prior uncertainties as a baseline and compare the posterior uncertainties from their two experiments.

5) Section 3.3 self-sensitivity calculation – It is slightly counter-intuitive that the single JR-STATION tower that is located at ∼60N, 130E provides the same influence on the analyses as all the other set of JR-STATION towers that are clumped together between 60-90E. As per previous studies (Cardinali et al. 2004, Liu et al. 2009, Kim et al. 2014), typically there is a negative correlation between the self-sensitivities and the spatial density of the observations. Can the authors comment on why that one single tower observation does not provide higher influence than a cluster of towers together?

6) Minor comments- (a) Kindly check the spelling of 'Eurasian Boreal' in Figure 5A. (b) The color scale for Figure 7 should be modified – either a linear increase or use something analogous to a log scale. Currently it jumps from a scale of 34-36 to 70-75. (c) For Section 3.4 and Table 5, the authors should choose a set of studies spanning the same spatial domain, temporal extent, space-time resolution at which fluxes are estimated and then compare to their estimates from the CNTL and the JR experiments. This would help out bring out the main message in this section.

---

## Referee Comment (RC2) · Anonymous Referee #1 · 1 Jun 2016

Review Kim et al. ACPD 2016

General comments

Kim et al.'s paper discusses the additional constraints on net biogenic CO2 fluxes in Siberia brought by adding specific regional CO2 observation. The authors add on top of a global ('control') data set (NOAA, WDCGG) additional data that is the Japanese Russian ('JR') network of station in Siberia. The control data set lacks stations over Siberia, a gap that the JR network fills successfully. The inversion set up is the well-established CarbonTracker (CT), and inversions of the control and control+JR sites are analysed comparatively. This paper discuss that adding JR observations in the setup, in the vast, poorly-sampled region of Siberia, brings additional information when estimating top-down fluxes. The CT inversion set up used in the present paper was

described in two other recent papers (Kim et al. 2014a, b), where the authors applied CarbonTracker with a focus in Asia (including Siberia) to the 'control' data set. Previously, Saeki et al. (2013) already evaluated the impact of adding this same JR data set on the mean biogenic CO2 flux, albeit with a different inversion set-up. Kim et al. find in relative terms a similar reduction in flux uncertainty when adding the JR network in the inversion. The paper lacks sufficient discussion on the ability of their model to reproduce JR and independent observations in Siberia, and rely on high level statistical analysis instead, which limits a deeper understanding of the problem. The paper also lacks distance to the opportunities and limitations associated to inverse modelling in an 'under-documented' region such as Siberia. However the material at hand is very valuable and provides potentially a basis for an in-depth discussion of CO2 fluxes over Siberia. Therefore I suggest rejecting the manuscript to allow its authors to improve it and eventually resubmit.

Detailed comments

Abstract Please specify in the abstract the time period over which the study is done. The abstract should be more quantitative about the fluxes to illustrate the improvement brought by the additional data, in terms of control, updated fluxes, and uncertainy reductions. The abstract could state the improvements obtained over Saeki et al. (2013). P2 L6 'useful information': please provide a quantitative estimate for this statement. Last sentence: please also mention the contribution to the estimation of European fluxes.

Introduction The first paragraph should mention also comparisons with inversion results, e.g. Dolman et al., Biogeosciences, 9, 5323-5340, 2012 The second paragraph (p2 l20-21) should discuss other factors leading to error in inverse modelling results, e.g. model error, representation error etc.: data sparseness is not the only one. P3 l5 after last sentence please discuss results from previous research, including Saeki et al. , 2013 here, as well as Dolman et al. 2012 (see above), and Berchet et al. (Biogeosciences, 12, 5393–5414, 2015 ).

P4 l22: 'CO2 uptake (. . .) slightly increase': compared to what and by how much? It could be useful for the reader to be reminded if Zhang et al. used (landing/take off) vertical profiles or (cruise altitude) tropopause data.

Methodology Here the section is written for readers already initiated in CarbonTracker (CT). Many items are not explained or assuming a detailed prior knowledge of CT.

P5 l9: 'emissions': could these F's be defined as the 'a priori' emissions?

P5 l12 What is an 'ecoregion'?

P5 l13: Gurney et al (2002) uses 11 land regions and this research uses 126 land regions. Therefore the reference to Gurney 2002 might not be appropriate, or explanation lacking. Please explain the difference in region definition.

P5 l14-15: scaling factor: how are these (5 weeks, 1 week) durations chosen?

P5 l15: What is an assimilation cycle in this context?

P5 l15-17 the two sentences ('In each assimilation. . . assimilation cycle.') are unclear, please revise the explanation.

P7 l7 is EDGAR corrected for interannual growth of CO2?

P7 l22-24: Is there a correction applied to account for the difference of the NIES scale in the inversion? The paper should mention how does this bias translates into uncertainty (especially when inversion correction modifies the balances Siberia vs Europe)

P8 l4: regarding the notion of having 'the same' MDM: it is not consistent with the fact that MDM is specified above to be determined by Eq. 6 in the paper. How can it be equal to 3 ppm? The authors should also explain why the same confidence is given to the JR-STATION network MDM (3 ppm) as for the US network, given their different tower design (e.g. sampling height) . The ability of models to reproduce the observations needs to be discussed to comfort the chosen value.

[Figure]

P9 l4 The difference between the inversions should also include a comparison to the prior fluxes for the sake of further discussion. 'The difference in fluxes . . . distinctive in . . . Siberia': Here it seems from Fig 2 that the fluxes are modified because the CNTL inversion puts an anomalous large sink in Siberia in the absence of local measurements. Therefore the JR run brings back fluxes to a realistic value. The difference is reflecting a particular approach (be it prior fluxes or optimization process) chosen by the inversion.

P9 l13: How is the 1 sigma standard deviation determined (on which basis: ensembles, . . .)? Does error on prior fluxes intervene in the inversion uncertainty?

P9 l27: 'uncertainty of the. . . uptake. . . is reduced': this is quite expected and I suggest adding the word 'is expectedly reduced'. However in terms of relative uncertainty (error ratio to estimated fluxes) no progress has been made. This reflects the fact that the CNTL inversion tends to allocate strangely large fluxes to Siberia in the absence of atmospheric measurement able to constrain this region

P10 l2-3: Since the drought affects Europe to a large extent, and the dataset is not different over Europe for the two runs JR and CNTL, it is not plausible that the drought can be used as an explanation for differences between JR and CNTL runs. Please revise this part.

P10 l5: This difference of number of observation and its impact on the fluxes may partly explain the time pattern of Fig. 4. Please provide some quantified information on the impact of the evolution of observations on the time pattern ( e.g. by removing some sites or maintaining a sparse network and comparing with the long term run).

Explanations for the trends observed in Fig 4 and 5 should be discussed, see e.g. Sitch et al (2015, Biogeosciences, 12, 653–679)

P11 l16: what are background surface co2 fluxes?

P11 l18-20: How are the results of this study sensitive to MDM? could the authors test

this assumption (well prescribed MDM) with different MDM values? Inversely, do poor (different from 1) values of this chi-2 parameter for other stations imply that the MDM is poorly prescribed? (e.g. BAL, MNM, . . .).Is MDM not dependent on sampling height in the JR station network?

To support this statement the authors should show and discuss comparison of JR-station $CO_2$ observations and model (prior and optimized, in the JR experiment context).

P11 l29. Sites with 7.5 ppm MDM are presented as afflicted by poor model simulation of their observed $CO_2$. However no confidence is given about the JR-STATION sites in terms of the accuracy of model representation of $CO_2$ at these sites, only a mean bias which is a very limited measure of model—observation fit improvement. This should be presented and extensively discussed prior to discussing the result of inverse modelling with a blind approach to the forward simulated $CO_2$ (this is also directly related to the comment above).

Overall section 3.2 is too limited and lacks a conclusion to support the subsequent analyses, especially in view of demonstrating the value of additional observations offered by JR-STATION.

In section 3.3. Fig 7 8 and 9 are valuable but not sufficient in themselves to allow the reader to appreciate the contribution of the additional JR observations. a mapping of prior uncertainties, CNTL UR and JR UR would be required to support the discussion.

P12 l15: 'additional observations sometimes have a great impact' : Please be more explicit and quantitative about 'sometimes' and 'great impact'.

P12 l16: The author find stronger UR in summer than in winter. Is this due to a higher uncertainty related to larger net fluxes in summer relative to winter? How is this seasonal UR difference explained over Siberia?

P13 l7-8: please give more details about why high 1-week RMSD and self sensitivities

of JR STATION sites is consistent.

P13 l9: 'it takes 5 weeks to affect the surface CO2 fluxes in Siberia by the transport of CO2 concentrations' : this statement is not supported by the demonstration in Kim et al. (2014, see their Fig 13), who only compared 1 week and 5 weeks, but not other time intervals. Therefore this incorrect interpretation needs to be reformulated. I could suggest a sentence such as 'it takes more than 1 week to affect the surface CO2. . .', which is better supported by the elements provided.

However this observation by the authors is important. If the correction of Siberian surface fluxes, in CNTL, is only performed based on air masses between 1 and 5 weeks, it means that Siberia is an underconstrained region in terms of CO2 fluxes (and this is a conservative statement). As a result, comparisons between CNTL and JR should take into account the large 'weakness' of fluxes allocated to Siberia by the CNTL inversion. This is illustrated in Table 2: EB is the region with the strongest increment between a priori and CNTL (from -0.07 to -1.17 PgC/y), but at the same time it is the only region with the least in situ observations. The other regions have smaller increment, and at the same time more 'local' observations are available. This bias needs to be explicited and discussed. The sentence of the abstract (p2, l1) comparing the fluxes calculated with the additional observations to the fluxes calculated without, suggests as such that the two flux values can be compared directly ('uptake. . . decreased'). On the contrary, Siberian fluxes calculated without the additional JR observations are highly sensitive to many assumptions within the inversion, and therefore any direct comparison requires a clear statement that this comparative approach is dependent on the inversion set up. This is also true for the sentence concluding this section (p13 l14-16).

Section 3.4. This section should also propose comparison with inversions intercomparison excercises, such as the TRANSCOM intercomparisons, see Gurney et al. (2002) or RECCAP Peylin et al (2013 Biogeosciences, 10, 6699-6720, doi:10.5194/bg-10-6699-2013). Please also compare with the synthesis work of Dolman et al. (2012)

[Figure]

P13 l26 the paragraph is concluded by the importance of the inversion framework used. Therefore the difference with the results of Saeki et al. (2013) needs to be examined in more detail. This paragraph leads the reader to the obvious conclusion that the choice of the inversion setup has as much impact on the posterior fluxes uncertainty of Siberia than the addition of a novel observation network such as JR-STATION.

Comparing the study reviewed and Saeki et al. (2013) from the numbers provided in the papers, it is striking that the two studies consistently conclude, in relative terms, both to a Siberian flux that is lower by about one third when adding JR-STATION (-37.5% for Saeki et al., -34.1% for this study), and decrease their uncertainty by about one-quarter (-22.7% and -24.7% respectively). At the same times, when comparing the two studies (being based on similar observation data sets), Kim et al.'s find fluxes consistently two times higher that Saeki et al. with or without the JR-STATION dataset, and report uncertainty that are 15% higher, with or without the JR STATION dataset.

Therefore the reported numbers lead to the observations that a change in inversion setup has more impact on the estimated fluxes in Siberia than using or not the only existing dataset in the region. This is stimulating because it means that more research is needed before $CO_2$ budgets calculated using inverse methods over Siberia can be reliably used. It supports the suggestion for extensive comparison of the simulated and measured $CO_2$ at the JR STATION sites in this study.

The numbers above also imply that the current set up questions the constraints on $CO_2$ fluxes (in terms of range of likely flux value) over Siberia reported by Saeki et al. 2013, and this requires further comparison with other studies, possibly bottom up allometric or modelling studies. It should be noted also that Maki et al. 2010 reported smaller uncertainty (0.41 PgC/y) , even without the JR-STATION network. This requires a detailed comparison of these studies for the sake of coherence.

P14 l2 and 3: what is the uncertainty range of single-year flux values (here 2008 and 2009 are discussed)?

[Figure]

Section 4 Please revise the conclusions according to the discussion above. What are the key challenges identified by the authors before estimating robust CO2 fluxes over Siberia from atmospheric inverse modelling?

P15 l9 is the longitudinal redistribution toward Europe a conclusion shared by Saeki et al., 2013?

P32 fig 5 further discussion of the trend in European fluxes and its possible drivers would be valuable.

P33 fig 6 please add the prior fluxes seasonal cycles to each panel

Editorial comments

There are several occurrences of unexplained acronyms; the authors should check this carefully.

P2 l13: typo: Schulze et al, not Schuleze et al

P3 l10: 'useful information': please be more specific on the usefulness of this information.

P3 l18: due to the difference in time period the word 'accompany' is not correct, should be e.g. 'preceded'

P3 l21: typo: YAK-AEROBO -> YAK-AEROSIB.

P3 l25 : multi-year Zotto measurements can certainly be used for inverse modelling.

P4 l11: 'increasing the sites': should be 'increasing the number of sites'

P5 l19: expand EnKF acronym

P5 l22 (eq. 2) and l26: please be consistent on notation: x (superscript) b or x (subscript) b

P6 l7-13: ensembles, perturbation, background error, localization, physical distance:

these notions have not been introduced before, please provide guidance for the reader to understand

P6 l15-17: please revise this sentence for clarity and syntax

P7 l7 please give references for CDIAC and EDGAR.

P7 l12: please provide link for ESRL data set, and give credit consistently to organization operating the measurements (e.g. in Europe).

P7 l 29: Is the MDM 'determined' or incremented? It is unclear with this formulation. Please revise accordingly. What is intended by 'innovation' chi-2?

P8 l17: typo: exists -> exist

P8 l25: 'from two experiments': I suggest to change to a determined form 'from the two experiments'

P9 l4 'greater': please quantify in the text

P9 l14: global total optimized $CO_2$ fluxes': the wording is problematic because this does not include fossil fuel and forest fire. Should be e.g. 'total biogenic'.

P9 l20: typo 'between two the experiments' -> 'between the two experiments'

P10 l2 please revise and clarify: 'is reduced all years' , I suggest 'is reduced in JR for all years'

P10 l9 readability would be improve to write Siberia instead of 'EB'. What is 'ET'?

P10 l14 the figure is not a histogram (binned distribution) but a time series. Please correct accordingly.

P11 l1 'Additional Siberian data": very indeterminate formulation. Please add e.g. 'These additional JR Siberian data'

P11 l17 what are background scaling factor? This seemingly important notion should

be explained in the section on experimental framework.

P11 l22 ET: explain acronym

Section 3.3. : Sect. 3.3. or part thereof should be before 3.1 and 3.2 as these sections 3.1 and 3.2 discuss already on the basis of CNTL vs JR runs. The structure of the paper could be reassessed for the benefit of readability.

P12 l10: Conifer: typo (confer). Why are only Conifer Forest of EB mentioned with no discussion of other ecosystems? What do they represent vs the rest of the Siberian ecosystems?

P12 l11 'which has additional information'; I suggest to use a determinate form 'which has the additional information'

P12 l14: 'the magnitude of the maximum uncertainty reduction is higher than the average value': this is certainly trivial. Please remove this sentence. I don't see the value of maximum weekly UR at all and I suggest to remove this panel 7b

P12 l26 please relate the definition of self sensitivity to the

P12 l26 what are 'Continuous site category observations'?

P13 l6-7: 'fluxes. . . are analysed by direct observations at the first cycle': this sentence might not be clear. Please rephrase.

P13 l21 What is CT2013B?

P13 l22 Please be careful when reporting numbers. The uncertainty of 0.41 PgC/y is wrong , the correct number is given in your Table 5 (0.61).

P15 l20. Please add acknowledgments for other observational data providers.

P23 Table 1 the table needs to differentiate altitude and sampling height, which is a potential indicator of how difficult it is to simulate a particular site. Also please make sure the proper credits are given to the providing Laboratories (last 7 lines, in Europe).

P24 Table 2 I suggest to add total Northern hemisphere

P27 CT2013B and CTE2014: please give reference.

P32 Fig 5 Panel (a): please correct typo (Euraisan -> Eurasian)

P32 l3 there are no 'ocean' in this figure, please correct Fig 5 caption accordingly.

---

## Author Comment (AC1) · 26 Aug 2016

ACP-2015-875 (Editor – William Lahoz)

Response to Reviewer 2

The authors thank the reviewer 2 for a thoughtful review of the manuscript. We agree with the reviewer's points and have made the necessary changes. The responses for the reviewer's specific comments are as follows.

**Comment:**

*This study evaluates the influence of additional CO2 observations (from the JRSTATION towers) on the analysis of Eurasian and global CO2 surface fluxes. The novelty of this study is in using these additional tower observations, which have not been used within an inverse modeling study before. The results demonstrate that these observations do have a certain amount of impact, namely it adjusts the flux patterns and magnitudes between Eurasian Boreal (local) and other NH land regions (non-local). This is expected based on the way an inverse modeling system works, especially the resultant interplay between the observation density/network and the prior weighed by their respective covariances. This is not a novel finding in itself. What is of greater interest, are the adjustments that are made to the surface fluxes and whether they are correct or not (especially the reduced sink in Eurasian Boreal region). No independent evaluation, either of the posterior CO2 concentrations or the posterior fluxes with any kind of independent data has been provided, however. The authors have compared their flux estimates to a suite of previous studies. But these studies cover different temporal extent (i.e., span across a wide variety of years), and second all of the estimates fall within the reported uncertainty bounds. There is no rationale behind the authors claim that the flux estimates from the JR experiment are more comparable to the previous studies than the CNTL experiment (Page 14, Lines 4-5). It is also highly misleading that in Section 3.2 the authors show results comparing the posterior CO2 concentrations to the observations that are being assimilated in the first place. Finally in Section 3.3, the uncertainty reduction should be calculated individually for the CNTL and the JR experiments relative to the prior uncertainties that were specified. It is again misleading to compare two posterior uncertainties (without knowing which one provides a baseline) and call this calculation as an "uncertainty reduction".*

*The following points provide a checklist on critical sensitivity tests/issues that should be addressed to first validate the results presented in this study, and thereby make it relevant and appealing to the* carbon *science community.*

**Author's response**: Following the reviewer's suggestions, we have revised the manuscript substantially. Based on in-depth analysis of the two experiments, we have tried to show the ability of CarbonTracker to reproduce JR and other observations in Siberia by assimilating the additional JR station data. Specific responses to the reviewer's

comments and revisions are shown below.

**Specific Comment:**

*1) Evaluation of posterior CO2 concentrations with independent data – This is the most important step that is missing from this study. This should be done either by comparison with independent data or via data denial experiments. In the latter case, specific set of in situ observations that are common to both CNTL and JR experiments may be held in reserve (i.e., those data should not be assimilated into the CT system). The posterior CO$_2$ concentrations from the two experiments should be compared to this independent data both qualitatively and quantitatively.*

**Author's response**: Following the reviewer's opinion, we evaluated the posterior CO$_2$ concentrations from the two experiments with independent data. We used the airborne observations over BRZ (Berezorechka; 56.15°N, 84.33°E) in the taiga region of West Siberia (detailed explanation in Section 2.3) as the independent data. The results show that the optimized fluxes of JR experiment exhibit better agreement with independent observations in terms of root mean square difference, mean absolute error, and Pearson's correlation coefficient at all altitudes, which supports the usefulness of Siberian tower measurements on the estimation of surface CO$_2$ fluxes over Siberia. Table 5 and discussion of the results (Section 3.2) are added in the revised manuscript as follows.

Table 5. Bias, root mean square difference, mean absolute error, and Pearson's Correlation Coefficient of the model CO$_2$ concentration of CNTL and JR experiments in comparison with the vertical profile of CO$_2$ concentrations at BRZ site.

| Altitude (km) | Bias (ppm) | | Root-Mean-Square Difference (ppm) | | Mean Absolute Error (ppm) | | Pearson's Correlation Coefficient | |
|---|---|---|---|---|---|---|---|---|
| | CNTL | JR | CNTL | JR | CNTL | JR | CNTL | JR |
| ~ 0.5 | -0.13±4.81 | 0.20±4.57 | 4.82 | 4.57 | 3.45 | 3.23 | 0.95 | 0.95 |
| 0.5 ~ 1.0 | 0.58±4.30 | 0.83±4.10 | 4.34 | 4.18 | 3.14 | 3.03 | 0.95 | 0.95 |
| 1.0 ~ 1.5 | 0.40±3.94 | 0.56±3.69 | 3.96 | 3.74 | 2.88 | 2.68 | 0.93 | 0.94 |
| 1.5 ~ 2.0 | 0.25±3.46 | 0.42±3.24 | 3.47 | 3.27 | 2.49 | 2.34 | 0.93 | 0.94 |
| 2.0 ~ 2.5 | 0.43±3.20 | 0.59±2.91 | 3.22 | 2.97 | 2.35 | 2.18 | 0.92 | 0.94 |
| 2.5 ~ 3.0 | 0.56±2.89 | 0.73±2.58 | 2.94 | 2.69 | 2.21 | 2.08 | 0.90 | 0.92 |
| 3.0 ~ | 0.13±3.19 | 0.44±2.65 | 3.19 | 2.68 | 3.89 | 2.03 | 0.86 | 0.90 |

We have revised the sentences in Section 2.3 as follows.

"(3) JR-STATION observation data over Siberia operated by CGER/NIES (Sasakawa et al., 2010; 2013). The JR-STATION sites consist of nine towers (eight towers in west

Siberia and one tower in east Siberia). Atmospheric air was sampled at four levels on the BRZ tower and at two levels on the other eight towers. At the BRZ (Berezorechka) site in west Siberia, both tower and aircraft measurements are sampled. The light aircraft at BRZ site measures the vertical profiles of $CO_2$ from the PBL to the lower free troposphere and these vertical profiles are used as independent observations for verification."

We have added the following sentences at the end of Section 3.2.

"In addition, model $CO_2$ concentrations calculated by optimized fluxes of the two experiments are compared with independent, not assimilated, vertical profiles of $CO_2$ concentration measurements by aircraft at BRZ site in Siberia. Table 5 presents the average bias, root-mean-square difference (RMSD), mean absolute error (MAE), and Pearson's correlation coefficient of the model $CO_2$ concentrations calculated by optimized fluxes of the two experiments based on the observations at BRZ site as the reference. The statistics are calculated at each vertical bin with 500 meter interval. Overall, the biases of two experiments are less than 0.83 ppm showing good consistency between model and observed $CO_2$ concentrations. The biases of the CNTL experiment are smaller than those of the JR experiment at all altitudes, whereas the standard deviations of the CNTL experiment are greater than those of JR experiment, which implies that the biases of the CNTL experiment fluctuate as its average more than those of the JR experiment. In contrast, the RMSD and MAE of the JR experiment are smaller than those of the CNTL experiment, and the correlation coefficient of the JR experiment is greater than that of the CNTL experiments. Therefore, overall the statistics show that the model $CO_2$ concentrations of the JR experiment is relatively more consistent with independent $CO_2$ concentration observations compared to those of the CNTL experiment over Siberia. "

*2) Uncertainty estimates associated with the analyzed flux estimates – On Page 9, Lines 13-14, the authors claim that the ". . .uncertainty is calculated as one-sigma standard deviation of the fluxes estimated, using Gaussian errors". It is unclear why the authors choose this approach when they are using an ensemble Kalman filter based system, where they should be able to directly recover the posterior uncertainty over the entire time period. Why is such an ad-hoc approach used to calculate the uncertainty? What is the basis for this approach?*

**Author's response**: The uncertainty estimation in this study is not based on ad-hoc method but based on uncertainty estimation method used in previous studies using CarbonTracker (Peters et al., 2007, 2010; Zhang et al., 2014a, 2014b). As mentioned by the reviewer, in Ensemble Kalman Filter system, the posterior uncertainty can be estimated directly from the ensembles. One sigma standard deviation of surface fluxes was calculated based on ensembles of prior and optimized surface fluxes. To clarify the uncertainty estimation method, we have revised the manuscript as follows.

"The difference in the optimized CO₂ flux between the two experiments is analyzed. Table 2 presents prior and optimized fluxes with their uncertainties for global total, global land, global ocean, NH total, Tropics total, Southern Hemisphere total, and TransCom regions in the NH. Flux uncertainties are calculated from the ensembles of prior and optimized surface fluxes assuming Gaussian errors, following previous method used in Peters et al. (2007, 2010)."

*3) Reduced uptake estimated in EB between 2002-2009 – Possibly the real significant finding from the additional JR-STATION tower observations are that the overall magnitude of the uptake reduced in Eurasian boreal region during NH summer. This is a reasonable conclusion for the summer of 2003 (anomalous drought for this year) but the authors claim a consistent reduction averaged out over the entire 2002-2009 period. The authors do not address any underlying mechanism for this difference in uptake from the two experiments. Is this simply an artifact of the inverse modeling setup, interplay between data density, error covariances, etc.? Or are there changes in vegetative activity that took place during this period in the Eurasian Boreal region and the JR-STATION tower observations were able to observe those local changes. The authors need to provide some form of mechanistic understanding for their inverse modeling results.*

**Author's response:** In 2003, the uptake in EB in the JR experiment was not reduced, but increased. The reason is that the drought in Europe affected the reduced uptake in EB in the CNTL experiment whereas the uptake in EB is actually not that much reduced.

The CNTL and JR experiments have the same inversion modeling setup except the observation data set (with or without JR-STATION data). Therefore, the differences in flux uptakes over Northern Hemisphere were from the additional JR-STATION data used in the inversion. The JR-STATION tower observations are able to observe those local vegetation activities in boreal summer appropriately as shown in Fig. 6. Without JR-STATION data, the surface flux uptakes are determined mostly by the transport model and remote observations. By adding JR-STATION data, the surface flux uptakes in Siberia are constrained both the model and JR-STATION observations. The differences between observed and model CO₂ concentrations simulated using optimized surface fluxes in JR experiment are much smaller than those in CNTL experiment at JR-STAION sites, which implies that an appropriate agreement between observed and optimized surface CO₂ concentrations over Siberia in JR experiment.

Therefore, the previous misleading texts in Section 3.1 was revised as follows.

"The uptake of optimized surface CO₂ flux in this region is reduced in JR for all years except 2003. In 2003, extreme drought occurred in the northern mid-latitudes (Knorr et al., 2007) and Europe (Ciais et al., 2005), which resulted in increased NEE (i.e. reduced uptake of CO₂) in EB in the CNTL experiment. The uptake of optimized surface CO₂ fluxes in Siberia in 2003 is reduced in the CNTL experiment due to the remote effect of

drought in Europe. Despite the number of JR-STATION data used in the optimization in 2003 being relatively smaller than that in the later experiment period, new observations in the JR experiment provide information on the increased uptake of optimized surface $CO_2$ fluxes in 2003 in Siberia (Fig. 3b)"

*4) Prior flux estimates and associated uncertainty used throughout the study – For Figures 4,5 and 6 the authors should add the prior flux estimates (say green or gray bars/lines) to the figures. For the uncertainty reduction reported in Section 3.3, the authors should use the prior uncertainties as a baseline and compare the posterior uncertainties from their two experiments.*

**Author's response**: Following the reviewer's opinion, prior fluxes are added in Figs. 4, 5, and 6 in the revised manuscript. In addition, we have added explanations and comparisons of prior and posterior fluxes accordingly in the revised manuscript.

We have plotted the uncertainty reductions (UR) of CNTL experiment and JR experiment from their prior uncertainties and the difference of two URs (Fig_rev 1). A mapping of prior uncertainties is not shown because the prior uncertainties do not show the contribution of the additional observations. Except the EB region (i.e. Siberia), the average URs of two experiments show similar patterns in the Northern Hemisphere. The difference between the URs of CNTL (Fig._rev 1a) and JR (Fig._rev 1b) is readily apparent in Siberia (Fig._rev 1c), which is very similar result with the UR using Eq. (7) shown in Fig. 7c. Because the Fig. 7 in the manuscript already shows the contribution of the additional JR observations clearly and the URs of CNTL and JR from the prior uncertainties do not provide additional information on the impact of Siberian observations, we did not insert Fig._rev 1 in the revised manuscript. Instead, we have added the texts to read,

"The uncertainty reduction of CNTL and JR experiments based on the prior uncertainty as the reference ($\sigma_{prior}$ used instead of $\sigma_{CNTL}$ in Eq. (8); or $\sigma_{CNTL}$ used instead of $\sigma_{JR}$ in Eq. (8)) shows similar values in the NH except in Siberia region (not shown). In addition, the difference between average uncertainty reduction of CNTL and JR experiments based on the prior unceatinty as the reference (not shown) is very similar to the average of uncertainty reduction in Eq. (8) shown in Fig.7a."

[Figure]

Figure_rev 1. Average uncertainty reduction (%), based on the prior uncertainty as a reference, of (a) CNTL experiment and (b) JR experiment. (c) The difference between (a) and (b).

*5) Section 3.3 self-sensitivity calculation – It is slightly counter-intuitive that the single JR-STATION tower that is located at 60N, 130E provides the same influence on the analyses as all the other set of JR-STATION towers that are clumped together between 60-90E. As per previous studies (Cardinali et al. 2004, Liu et al. 2009, Kim et al. 2014), typically there is a negative correlation between the self-sensitivities and the spatial density of the observations. Can the authors comment on why that one single tower observation does not provide higher influence than a cluster of towers together?*

**Author's response**: The average self-sensitivity of YAK site located in east Siberia is 10.8% which is the largest sensitivity value among the JR-STATION sites. The average of the self-sensitivities for other eight sites located between 60°E and 90°E is 8.4%. Therefore, YAK site provides greater impacts than a cluster of Siberian towers. This is

intuitive result. To clarify, we have revise text to read, "The average self-sensitivities of additional observations are higher than those of other sites, providing much information for estimating surface $CO_2$ fluxes. In particular, YAK site located in east Siberia provides greater impacts than other JR-STATION sites located in 60 ~ 90°E."

**Minor Comment:**

*(a) Kindly check the spelling of 'Eurasian Boreal' in Figure 5A.*

> **Author's response**: Following the reviewer's opinion, we have revised the manuscript.

*(b) The color scale for Figure 7 should be modified – either a linear increase or use something analogous to a log scale. Currently it jumps from a scale of 34-36 to 70-75.*

> **Author's response**: Following the reviewer's opinion, the color scale for Fig. 7 was modified. In addition, following the other reviewer's opinion, Fig. 7b was removed in the revised manuscript.

*(c) For Section 3.4 and Table 5, the authors should choose a set of studies spanning the same spatial domain, temporal extent, space-time resolution at which fluxes are estimated and then compare to their estimates from the CNTL and the JR experiments. This would help out bring out the main message in this section.*

> **Author's response**: We agree with the reviewer's opinion, choosing a set of studies spanning the same spatial domain, temporal extent, space-time resolution is important in comparing this study with other studies. We have chosen the same spatial domain (Eurasian Boreal and Europe) with other studies. However, it is difficult to match the same temporal extent because each study use different experimental period. For example, Saeki et al. (2013; Table 6) and Zhang et al, (2014b; Table 7) compared their estimated fluxes with those of other studies for different time periods.

> We have tried to match the temporal period between this study and other CT framework results that are provided in each CarbonTracker's homepage. Therefore, Table 6 is partially revised as follows.

[revised manuscript text omitted]

---

## Author Comment (AC2) · 26 Aug 2016

ACP-2015-875 (Editor – William Lahoz)

Response to Reviewer 1

The authors thank the reviewer 1 for a thoughtful review of the manuscript. The responses for the reviewer's specific comments are as follows.

**General Comment:**

*Kim et al.'s paper discusses the additional constraints on net biogenic CO2 fluxes in Siberia brought by adding specific regional CO2 observation. The authors add on top of a global ('control') data set (NOAA, WDCGG) additional data that is the Japanese Russian ('JR') network of station in Siberia. The control data set lacks stations over Siberia, a gap that the JR network fills successfully. The inversion set up is the well established CarbonTracker (CT), and inversions of the control and control+JR sites are analysed comparatively. This paper discuss that adding JR observations in the setup, in the vast, poorly-sampled region of Siberia, brings additional information when estimating top-down fluxes. The CT inversion set up used in the present paper was described in two other recent papers (Kim et al. 2014a, b), where the authors applied CarbonTracker with a focus in Asia (including Siberia) to the 'control' data set. Previously, Saeki et al. (2013) already evaluated the impact of adding this same JR data set on the mean biogenic CO2 flux, albeit with a different inversion set-up. Kim et al. find in relative terms a similar reduction in flux uncertainty when adding the JR network in the inversion. The paper lacks sufficient discussion on the ability of their model to reproduce JR and independent observations in Siberia, and rely on high level statistical analysis instead, which limits a deeper understanding of the problem. The paper also lacks distance to the opportunities and limitations associated to inverse modelling in an 'under-documented' region such as Siberia. However the material at hand is very valuable and provides potentially a basis for an in-depth discussion of CO2 fluxes over Siberia. Therefore I suggest rejecting the manuscript to allow its authors to improve it and eventually resubmit.*

> **Author's response**: Following the reviewer's suggestions, we have revised the manuscript substantially. Based on in-depth analysis of the two experiments, we have tried to show the ability of CarbonTracker to reproduce JR and other observations in Siberia. Specific responses to the reviewer's comments and revisions are shown below.

**Detailed Comments:**

*1) Abstract Please specify in the abstract the time period over which the study is done. The abstract should be more quantitative about the fluxes to illustrate the improvement brought by the additional data, in terms of control, updated fluxes, and uncertainy reductions. The*

*abstract could state the improvements obtained over Saeki et al. (2013).*

**Author's response**: Following the reviewer's opinion, we have revised the abstract as follows.

"To investigate the effect of additional $CO_2$ observations in the Siberia region on the Asian and global surface $CO_2$ flux analyses, two experiments using different observation dataset were performed for 2000-2009. One experiment was conducted using a data set that includes additional observations of Siberian tower measurements (Japan-Russia Siberian Tall Tower Inland Observation Network: JR-STATION), and the other experiment was conducted using a data set without the above additional observations. The results show that the global balance of the sources and sinks of surface $CO_2$ fluxes was maintained for both experiments with and without the additional observations. While the magnitude of the optimized surface $CO_2$ flux uptake and flux uncertainty in Siberia decreased from -1.17±0.93 Pg C $yr^{-1}$ to -0.77±0.70 Pg C $yr^{-1}$, the magnitude of the optimized surface $CO_2$ flux uptake in the other regions (e.g., Europe) of the Northern Hemisphere (NH) land increased for the experiment with the additional observations, which affect the longitudinal distribution of the total NH sinks. This change was mostly caused by changes in the magnitudes of surface $CO_2$ flux in June and July. ~"

We have not mentioned Saeki et al. (2013) because a reference is not normally stated in the abstract except when the study is a follow-up study of the reference. In addition, we didn't state Saeki et al. (2013) because we have used a different framework (inversion system, transport model, observations, etc.) from Saeki et al. (2013). Instead, we have added verifications using independent observations in Section 3.2.

*2) P2 L6 'useful information': please provide a quantitative estimate for this statement. Last sentence: please also mention the contribution to the estimation of European fluxes.*

**Author's response**: Following the reviewer's opinion, we have revised the abstract as follows.

"The observation impact measured by uncertainty reduction and self-sensitivity tests shows that additional observations provide useful information on the estimated surface $CO_2$ flux. The average uncertainty reduction of the Conifer Forest of EB is 29.1% and the average self-sensitivities at the JR-STATION sites are approximately 60% larger than those at the tower measurements in North America. It is expected that the Siberian observations play an important role in estimating surface $CO_2$ flux in the NH land (e.g., Siberia and Europe) in the future."

*3) Introduction The first paragraph should mention also comparisons with inversion results, e.g. Dolman et al., Biogeosciences, 9, 5323-5340, 2012 The second paragraph (p2 l20-21)*

*should discuss other factors leading to error in inverse modelling results, e.g. model error, representation error etc.: data sparseness is not the only one. P3 l5 after last sentence please discuss results from previous research, including Saeki et al. , 2013 here, as well as Dolman et al. 2012 (see above), and Berchet et al. (Biogeosciences, 12, 5393–5414, 2015 ).*

**Author's response**: Following the reviewer's opinion, we have revised the first paragraph in Section 1 to read, "The terrestrial ecosystem in the Northern Hemisphere (NH) plays an important role in the global carbon balance (Hayes et al., 2011; Le Quéré et al., 2015). Especially, Siberia is considered to be the one of the largest $CO_2$ uptake regions and reservoirs due to its forest area (Schulze et al., 1999; Houghton et al., 2007; Tarnocai et al., 2009; Kurganova et al., 2010; Schepaschenko et al., 2011) and its dynamics and interactions with the climate have global significance (Quegan et al., 2011). Therefore, it is important to accurately estimate the surface $CO_2$ fluxes in this region. For instance, Dolman et al. (2012) estimated terrestrial carbon budget of Russia, Ukraine, Belarus, and Kazakhstan using inventory-based, eddy covariance, and inversion methods and showed that the carbon budgets produced by three methods agree within their uncertainty bounds."

We have revised the second paragraph to read, "To estimate the surface $CO_2$ flux, atmospheric $CO_2$ inversion studies are conducted using atmospheric transport models and atmospheric $CO_2$ observations (Gurney et al., 2002; Peylin et al., 2013). However, prior emission, measurement error of observation, observation operator including model transport, and representative error affect the uncertainty of atmospheric inversion results (Engelen et al., 2002; Berchet et al., 2015a). Along these factors, large uncertainties remain in the estimated surface $CO_2$ fluxes due to the sparseness of current surface $CO_2$ measurements assimilated by inverse models (Peters et al., 2010; Bruhwiler et al., 2011)"

We have revised the sixth paragraph to read, "The Center for Global Environmental Research (CGER) of the National Institute for Environmental Studies (NIES) of Japan with the cooperation of the Russian Academy of Science (RAS) constructed a tower network called the Japan-Russia Siberian Tall Tower Inland Observation Network (JR-STATION) in 2002 to measure the continuous $CO_2$ and $CH_4$ concentrations (eight towers in central Siberia and one tower in eastern Siberia) (Sasakawa et al., 2010, 2013). The vertical profile of $CO_2$ concentrations from the planetary boundary layer (PBL) to the lower free troposphere is also measured by aircraft at one site of the JR-STATION sites (Sasakawa et al., 2010, 2013). Saeki et al. (2013) estimated the monthly surface $CO_2$ flux for 68 subcontinental regions by using the fixed-lag Kalman smoother and NIES-TM transport model with JR-STATION data. They reported that the inclusion of additional Siberian observation data has an impact on the inversion results showing larger interannual variability over northeastern Europe as well as Siberia, and reduces the uncertainty of surface $CO_2$ uptake. Meanwhile, Berchet et al. (2015b) estimated regional $CH_4$ fluxes over Siberia in 2010 by using JR-STATION data."

The results of references suggested by the reviewer are mentioned in appropriate places

in Section 1, for a smooth flow of Introduction. Saeki et al. (2013) is already discussed in P3 L32 in the originally submitted manuscript. Doman et al. (2012)'s results are added in first paragraph and Berchet et al. (2015b) is added in sixth paragraph in Section 1.

*4) P4 l22: 'CO2 uptake (. . .) slightly increase': compared to what and by how much? It could be useful for the reader to be reminded if Zhang et al. used (landing/take off) vertical profiles or (cruise altitude) tropopause data.*

**Author's response**: To clarify, we have revised the texts as follows.

"The CONTRAIL measurements include ascending/descending vertical profiles and cruise data below tropopause. The results show that surface $CO_2$ uptake over the Eurasian Boreal (EB) region slightly increases from -0.96 Pg C $yr^{-1}$ to -1.02 Pg C $yr^{-1}$ for the period 2006-2010 when aircraft $CO_2$ measurements were assimilated. However, the surface measurements data over the EB region are still not used in the study by Zhang et al. (2014b)."

*5) Methodology Here the section is written for readers already initiated in CarbonTracker (CT). Many items are not explained or assuming a detailed prior knowledge of CT.*

**Author's response**: The main purpose of this study is to estimate the impact of additional observations (JR-STATION data) in Siberia on the optimized surface $CO_2$ fluxes over Eurasian Boreal region as well as Northern hemisphere. Therefore, the methodology part of the manuscript contains an essential knowledge of CT necessary for understanding inversion results. Many previous studies using CT exist and explain much details about CT framework and methodology (Peters et al. 2007, 2010; Masarie et al. 2011; Kim et al. 2012, 2014a, 2014b; Zhang et al. 2014a, 2014b; Babenhauserheide et al. 2015; van der Laan-Luijkx et al. 2015). Therefore, we have moved the text to read, "The detailed algorithm of inversion method used in this study can be found in Peters et al. (2007) and Kim et al. (2014a)." at the end of Section 2.1.

*6) P5 l9: 'emissions': could these F's be defined as the 'a priori' emissions?*

**Author's response**: As indicated, F's are a priori emissions. We have revised the manuscript as follows.

"where $F_{bio}(x,y,t)$, $F_{ocn}(x,y,t)$, $F_{ff}(x,y,t)$ and $F_{fire}(x,y,t)$ are a priori emissions from the biosphere, the ocean, fossil fuel, and fires. $\lambda_r$ is the scaling factor to be optimized in the data assimilation process,"

*7) P5 l12 What is an 'ecoregion'?*

**Author's response**: According to CarbonTracker documentation, the ecoregions represent large areas of land in a continent, which have similar ecosystem types, and are used to divide continents into smaller pieces for optimization. To avoid the confusion in the original text, we have revised the text to read, "$\lambda_r$ is the scaling factor to be optimized in the data assimilation process, corresponding to 156 regions around the globe ~".

*8) P5 l13: Gurney et al (2002) uses 11 land regions and this research uses 126 land regions. Therefore the reference to Gurney 2002 might not be appropriate, or explanation lacking. Please explain the difference in region definition.*

**Author's response**: Following the reviewer's opinion, we have revised the manuscript for clarification as follows.

"In the land, the ecoregions are defined as the combination of 11 land region of Transcom regions (Gurney et al., 2002) with 19 land-surface characterization based on Olson et al. (1992). Inappropriate combinations of TransCom regions and Olson types are excluded. In the ocean, 30 ocean regions are defined following Jacobson et al. (2007)."

*9) P5 l14-15: scaling factor: how are these (5 weeks, 1 week) durations chosen?*

**Author's response**: The assimilation window (5 weeks, 1 week) used in this study follows previous studies for CarbonTracker (e.g., Peters et al., 2007; 2010, Kim et al., 2012, 2014a, 2014b, Zhang et al., 2014a, 2014b; van der Laan-Luijkx et al., 2015). The previous studies have shown that this configuration is appropriate to estimate the surface $CO_2$ flux using CarbonTracker. We have revised the manuscript as follows.

"The scaling factor spans 5 weeks with 1 week resolution. Several previous studies for CarbonTracker (e.g., Peters et al., 2007; 2010, Kim et al., 2012, 2014a, 2014b, Zhang et al., 2014a, 2014b; van der Laan-Luijkx et al., 2015) showed that 5 weeks of lag and 1-week time resolution are appropriate for optimizing the surface $CO_2$ fluxes.".

*10) P5 l15: What is an assimilation cycle in this context?*

**Author's response**: Assimilation cycle means an analysis step in the inversion process. Optimization of surface $CO_2$ fluxes is performed every assimilation cycle. We have revised the texts to read, "In each assimilation cycle (i.e., analysis step), ~ ".

*11) P5 l15-17 the two sentences ('In each assimilation. . . assimilation cycle.') are unclear, please revise the explanation.*

**Author's response**: Following the reviewer's suggestion, we have revised the text to read, "In each assimilation cycle (i.e., analysis step), the entire scaling factor for 5 weeks is updated by 1 week observations measured most recent week by a time stepping approach. The smoother window moves forward by 1 week at each assimilation cycle. After 5 assimilation cycles, the first part of the scaling factor analyzed by 5 weeks observations is regarded as the optimized scaling factor. The more detailed information of the assimilation process can be found in Kim et al. (2014b). ".

Because a schematic diagram of the assimilation process is already shown in Fig. 1 of Kim et al. (2014b) below, the interested reader is diected to Kim et al. (2014a) for more information about the assimilation process used in CT.

[Figure]

Figure 1. Schematic diagram of the assimilation process employed in CarbonTracker. In each analysis cycle, observations made within one week are used to update the state vectors with a five-week lag. The dashed line indicates how the simple dynamic model uses analysis state vectors from the previous one and two weeks to produce a new background state vector for the current analysis time. The TM5 model is used as the observation operator to calculate the model $CO_2$ concentration for each corresponding observation location and time (Courtesy of Kim et al. 2014b).

*12) P7 l7 is EDGAR corrected for interannual growth of CO2?*

**Author's response**: In CarbonTrakcer, annual global total fossil fuel $CO_2$ emissions are from the CDIAC. EDGAR provides annual emission estimates at 1°×1° resolution. Fluxes are spatially distributed in two steps: First, the coarse scale flux distribution country totals from CDIAC are mapped onto a 1°x1° grid; next, the country totals within the countries are distributed according to the spatial patterns from the EDGAR inventories (Documentation CT2010). Time series of global fossil fuel emission used in this study is shown in Figure_rev 1.

[Figure]

Figure_rev 1. Times series of global fossil fuel emission. (Courtesy of Documentation CT2010 [available online at: http://www.esrl.noaa.gov/gmd/ccgg/carbontracker/CT2010/ documentation_CT2010.pdf]).

To clarify, we have revised the texts to read, "~ (4) the prescribed fossil fuel emissions are from the Carbon Dioxide Information and Analysis Center (CDIAC, Boden et al., 2010) and the Emission Database for Global Atmospheric Research (EDGAR, European Commission, 2009) databases. The annual global total fossil fuel emissions are based on CDIAC. Fluxes at 1°x1° resolution are spatially distributed according to the EDGAR inventories."

*13) P7 l22-24: Is there a correction applied to account for the difference of the NIES scale in the inversion? The paper should mention how does this bias translates into uncertainty (especially when inversion correction modifies the balances Siberia vs Europe)*

**Author's response**: No correction was applied to the NIES scale in the inversion. As explained the manuscript, according to Machida et al. (2011), NIES 09 $CO_2$ scale is

lower than the WMO-X2007 $CO_2$ scale by 0.07 ppm at approximately 360 ppm and consistent in the range between 380 and 400 ppm. Most observations used in the assimilation are between 360-420 ppm (Fig_rev. 2). The assigned MDM of JR-STATION data is 3 ppm, which reflects the measurement error of observations and is much larger value than difference between NIES 09 $CO_2$ scale and WMO-X2007 $CO_2$ scale. In this assignment, the JR station data do not constrain the optimized flux more greatly than their confidence. Moreover, $CO_2$ measurements with the NIES scale have been successfully used in the many inversion studies. For example, Zhang et al. (2014b) assimilated Comprehensive Observation Network for Trace gases by Airliner (CONTRAIL) $CO_2$ observations in CarbonTracker. Saeki et al. (2013) used JR-STATION data with NOAA $CO_2$ data in the same inversion framework.

[Figure]

Figure_rev 2. Times series of JR-STATION $CO_2$ data used in this study. The daytime (12-16 LST) averaged data are used in the assimilation.

*14) P8 l4: regarding the notion of having 'the same' MDM: it is not consistent with the fact that MDM is specified above to be determined by Eq. 6 in the paper. How can it be equal to 3 ppm? The authors should also explain why the same confidence is given to the JR-STATION network MDM (3 ppm) as for the US network, given their different tower design (e.g. sampling height). The ability of models to reproduce the observations needs to be discussed to comfort the chosen value.*

**Author's response**: When new observations are assimilated to CarbonTracker, the MDM is assigned by the site categories of the new observations. After then, innovation $\chi^2$ statistics are gathered during the optimization to evaluate whether they are close to 1. The statistics near 1 implies that the assigned MDM value is appropriate.

We have followed the above process when determining the MDM of JR station data. In CarbonTracker, MDM of continuous tower measurements is assigned 3 ppm. Following the site categories, we assigned 3 ppm MDM for JR-STATION data. After then, we have verified whether the assignment is appropriate by calculating $\chi^2$ statistics which turns to be close to 1. The statistics are from 0.84 to 1.36 (Table 4) and the results are discussed in detail in Section 3.2. The statistics of JR station are closer to 1 compared to the sites in ET and Europe which use the MDM from the original CarbonTracker 2010 release.

Although the tower design of US network and JR stations are different, the same type of site categories use the same MDM because both of them are continuous tower measurements and their $\chi^2$ statistics show appropriate values.

The misleading texts were revised as follows.

"In CarbonTracker, model data mismatch (MDM, **R** in Eq. (7)) is assigned by site categories. The location of each observation site is represented in Fig. 1. The assigned MDM requires innovation $\chi^2$ statistics in Eq. (7) become close to one at each observation site (Peters et al. 2007).

$$\chi^2 = \frac{(\mathrm{y^o\text{-}\mathbf{H}x^b})^2}{\mathbf{HP^bH^T+R}}, \tag{7}$$

where $\mathrm{y^o\text{-}\mathbf{H}x^b}$ represent innovation. The site categories and MDM values are assigned the same value as in previous studies (Peters et al., 2007; Kim et al. 2014b; Zhang et al., 2014b): marine boundary layer (0.75 ppm), continental sites (2.5 ppm), mixed land/ocean and mountain sites (1.5 ppm), continuous sites (3.0 ppm), and difficult sites (7.5 ppm). Continuous site category is generally used for observations measured continuously. For the JR-STATION sites that have continuous tower measurements, the MDM is set to 3 ppm, which is the same as tower measurements in North America."

The feasibility of assigned MDM of JR-STATION was evaluated by Eq. (7), using innovation $\chi^2$-statistics, which shows that average value of each site are from 0.84 to 1.36 as explained in Section 3.2.

*15) P9 l4 The difference between the inversions should also include a comparison to the prior fluxes for the sake of further discussion. 'The difference in fluxes . . . distinctive in . . . Siberia': Here it seems from Fig 2 that the fluxes are modified because the CNTL inversion puts an anomalous large sink in Siberia in the absence of local measurements. Therefore the JR run brings back fluxes to a realistic value. The difference is reflecting a particular approach (be it prior fluxes or optimization process) chosen by the inversion.*

**Author's response:** As shown in Fig. 1, there are observations over East Asia although not enough number of observations over Siberia. The CNTL inversion results were produced by assimilating many observations over the globe including East Asia. The

observations over East Asia (located partly border of Siberia) affect the optimized fluxes over Siberia by 5-week-lag of the assimilation window. Thus large flux uptakes in Siberia were constrained by atmospheric $CO_2$ measurements that are located remotely. In addition, the large flux uptakes in Siberia have been noticed most of the CarbonTracker results as well as other inversion systems as reported in Peylin et al. (2013) (e.g. Fig. 5 and Fig. S8 in Peylin et al., 2013). Therefore the difference between the two experiments is not reflecting a particular approach used in this study.

In Fig. 2, the direct comparison among the prior, CNTL, and JR are shown. In addition, the aggregated prior and optimized surface $CO_2$ fluxes of CNTL and JR experiments over Eurasian Boreal (Siberia) region are -0.07±1.10 Pg C yr$^{-1}$, -1.17±0.93 Pg C yr$^{-1}$, and -0.77±0.70 Pg C yr$^{-1}$, respectively (Table 2). As seen in the above numbers, the difference of surface $CO_2$ fluxes between prior and JR is still large over Siberia region. Therefore, we have revised the text to read, "The difference in fluxes between the prior and JR experiment is large in EB (Figs. 2a, d) although smaller than that between the prior and CNTL experiment (Figs, 2a, c). The differences in fluxes between the CNTL and JR experiments are distinctive in EB (Siberia) where the new additional observations are assimilated (Fig. 2b)."

*16) P9 l13: How is the 1 sigma standard deviation determined (on which basis: ensembles, . . .)? Does error on prior fluxes intervene in the inversion uncertainty?*

**Author's response**: One sigma standard deviation of surface fluxes was calculated based on ensembles of surface fluxes. To clarify, we have revised the text to read, "Flux uncertainties are calculated from the ensembles of prior and optimized surface fluxes assuming Gaussian errors, following previous method used in Peters et al. (2007, 2010)."

Error on prior fluxes could intervene in the inversion uncertainty in the data assimilation process by means of the ensembles of prior surface fluxes which reflect the prior surface flux uncertainties and errors.

*17) P9 l27: 'uncertainty of the. . . uptake. . . is reduced': this is quite expected and I suggest adding the word 'is expectedly reduced'. However in terms of relative uncertainty (error ratio to estimated fluxes) no progress has been made. This reflects the fact that the CNTL inversion tends to allocate strangely large fluxes to Siberia in the absence of atmospheric measurement able to constrain this region*

**Author's response**: Following the reviewer's opinion, we have revised the texts to read, "The uncertainty of the optimized surface $CO_2$ uptake in the EB in the JR experiment is expectedly reduced by assimilating additional ~".

As shown in Fig. 1, there are observations over East Asia although not enough number of

observations over Siberia. The CNTL inversion results were produced by assimilating many observations over the globe including East Asia. The observations over East Asia (located partly border of Siberia) affect the optimized fluxes over Siberia by 5-week-lag of the assimilation window. Thus large flux uptakes in Siberia were constrained by atmospheric measurements located remotely. In addition, the large flux uptakes in Siberia have been noticed most of the CarbonTracker results. By assimilating additional observations over Siberia, the large flux uptakes in Siberia were reduced as shown in Fig. 7.

*18) P10 l2-3: Since the drought affects Europe to a large extent, and the dataset is not different over Europe for the two runs JR and CNTL, it is not plausible that the drought can be used as an explanation for differences between JR and CNTL runs. Please revise this part.*

**Author's response**: The drought over the northern mid-latitudes and Europe can affect the reduced uptakes in Siberia remotely through 5-week-lag of the assimilation window. Therefore, we think that the drought is associated with the reduced uptakes in EB (Fig. 5a) (Knorr et al., 2005). Because of remote drought effect, the flux uptakes in Siberia were reduced in CNTL. However, by assimilating observations over Siberia in JR, the flux uptakes were slightly increased. Therefore, the previous misleading text in Section 3.1 was revised as follows.

"The uptake of optimized surface $CO_2$ flux in this region is reduced in JR for all years except 2003. In 2003, extreme drought occurred in the northern mid-latitudes (Knorr et al., 2007) and Europe (Ciais et al., 2005), which resulted in increased NEE (i.e. reduced uptake of $CO_2$) in EB in the CNTL experiment. The uptake of optimized surface $CO_2$ fluxes in Siberia in 2003 is reduced in the CNTL experiment due to the remote effect of drought in Europe. Despite the number of JR-STATION data used in the optimization in 2003 being relatively smaller than that in the later experiment period, new observations in the JR experiment provide information on the increased uptake of optimized surface $CO_2$ fluxes in 2003 in Siberia (Fig. 3b)"

*19) P10 l5: This difference of number of observation and its impact on the fluxes may partly explain the time pattern of Fig. 4. Please provide some quantified information on the impact of the evolution of observations on the time pattern (e.g. by removing some sites or maintaining a sparse network and comparing with the long term run).*

**Author's response**: The texts "difference of number of observation" is for only JR-STATION data. Since 2002 when the JR-STATION site was started to operate at BRZ, the number of operation site and observations of JR-STATION have been increased. Therefore, the observation number of JR-STATION in 2003 is relatively small compared to that from 2004 to 2009.

To clarify the text, we have revised the texts to read, "Despite the number of JR-STATION data used in the optimization in 2003 being relatively smaller than that in the later experiment period, new observations in the JR experiment provide information on the increased uptake of optimized surface $CO_2$ fluxes in 2003 in Siberia."

Following the reviewer's opinion, we have added a discussion which may explain the time pattern of Fig. 5a. Although comprehensive observation system experiments such as data denial experiment or using different network provide quantitative information on the impact of the observation evolution on the time pattern of $CO_2$ fluxes, it is beyond the scope of this study. Besides, the similar patterns of CNTL and JR experiments in Fig. 4 imply that the impact of the evolution of observations on the time pattern may be small at least global sense. The local impact would be studied as a future work.

*20) Explanations for the trends observed in Fig 4 and 5 should be discussed, see e.g. Sitch et al (2015, Biogeosciences, 12, 653–679)*

**Author's response**: Following the reviewer's opinion, we have analyzed the trends observed in Figs. 4 and 5. We have revised the text as follows.

"Figure 4 shows the time series of annual and average prior and optimized surface $CO_2$ fluxes over global total, global land, and global ocean. For global total, the magnitude of optimized fluxes are much greater than that of prior fluxes due to the greater uptake of optimized fluxes than that of prior fluxes over global land (Figs. 4a and b). In contrast, the magnitude of optimized fluxes over global ocean is slightly weaker than that of prior fluxes (Fig. 4c). As shown in Table 2, the differences between annual and average optimized surface $CO_2$ fluxes over the globe are small and the average is almost the same for the two experiments (Fig. 4a) with a similar trend of -0.33 Pg C $yr^{-2}$ and -0.35 Pg C $yr^{-2}$ in CNTL and JR experiment respectively, and the differences in global land and ocean are also small (Figs. 4b, c) with a similar trend of -0.22 Pg C $yr^{-2}$ in global land of both CNTL and JR experiment and -0.11 Pg C $yr^{-2}$ and -0.13 Pg C $yr^{-2}$ in global ocean of CNTL and JR experiment respectively. The optimized surface $CO_2$ fluxes from each experiment show similar interannual variability, which implies that the additional Siberian observations do not affect the interannual variability of global surface $CO_2$ uptakes.

Figure 5 is the same as Fig. 4 but covers land regions in the NH. Although the optimized surface $CO_2$ fluxes over global total are similar, those over each TransCom region are different in each experiment. The optimized fluxes over each region show greater annual uptake relative to the prior fluxes in both experiment. The difference between the two experiments is largest in the EB as expected (Fig. 5a). The JR experiment exhibits a weaker surface $CO_2$ uptake in the EB than does the CNTL experiment except for 2003 as shown in Fig. 3b, whereas the JR experiment exhibits a greater surface $CO_2$ uptake in the other regions, especially over Europe in 2008 and 2009, than the CNTL experiment (Figs.

5b, c, d, and e). It is driven by the increase of $CO_2$ uptake in Eastern Europe (Figs. 3g and h). Because most of JR-STATION sites are located in the western part of Siberia (Fig. 1), the optimized surface $CO_2$ fluxes over Eastern Europe could be affected by JR-STATION observations. The trend of EB in CNTL experiment is -0.06 Pg C yr$^{-2}$, whereas that in JR experiment is 0.02 Pg C yr$^{-2}$ due to the reduced uptake of $CO_2$ in JR experiment since 2005 (Fig. 5a). As a result, the trends of the surface $CO_2$ uptake of EB and Europe in two experiments show opposite signs. In contrast, the surface $CO_2$ uptake trends of other land regions in NH are similar between the two experiments."

Below, the trends of surface $CO_2$ flux for each region (unit: Pg C yr$^{-2}$) are shown.

| Region | Experiment | |
|---|---|---|
| | CNTL | JR |
| Global total | -0.33 | -0.35 |
| Global land | -0.22 | -0.22 |
| Global ocean | -0.11 | -0.13 |
| Eurasian Boreal | -0.06 | 0.02 |
| Eurasian Temperate | -0.03 | -0.04 |
| North American Boreal | -0.03 | -0.04 |
| North American Temperate | -0.02 | -0.03 |
| Europe | -0.04 | -0.10 |

*21) P11 l16: what are background surface $CO_2$ fluxes?*

**Author's response**: Background surface $CO_2$ fluxes are calculated by multiplying background scaling factor (background state vector in Eq. (2)) to prior biosphere and ocean fluxes as in Eq. (1).

For readability, we have added the text to read, "The background surface $CO_2$ fluxes are calculated by multiplying the background scaling factor to prior biosphere and ocean fluxes as in Eq. (1).".

*22) P11 l18-20: How are the results of this study sensitive to MDM? Could the authors test this assumption (well prescribed MDM) with different MDM values? Inversely, do poor (different from 1) values of this chi-2 parameter for other stations imply that the MDM is poorly prescribed? (e.g. BAL, MNM, . . .). Is MDM not dependent on sampling height in the JR station network? To support this statement the authors should show and discuss comparison of JR station CO2 observations and model (prior and optimized, in the JR experiment context).*

**Author's response**: The results are sensitive to MDM. As answered in the reviewer's question 14 above, we have chosen 3 ppm MDM for JR-STATION data because they are continuous tower measurements same as the US network and their $\chi^2$ statistics showed appropriate values. The MDM of other stations (e.g. BAL, MNM, . . .) follows the

specification of CarbonTracker 2010 release. We cannot change the original specification of MDM for observation sites included in the public release of CarbonTracker. We guess that the MDM of other stations (e.g. BAL, MNM, . . .) was assigned that way because of poor model simulation of their observed $CO_2$ as indicated in manuscript. Same as other observation sites in CarbonTracker, the sampling height in the JR station was not considered in determining MDM.

We have shown and discussed the comparison of JR station $CO_2$ observations and model (prior and optimized) in the JR experiment context in Section 3.2 and Table 4. The 6th and 7th columns in Table 4 present the differences of background and optimized model from the observations in the JR experiment. In the JR experiments, the optimized model $CO_2$ concentrations are closer to observations compared to the background $CO_2$ concentrations.

*23) P11 l29. Sites with 7.5 ppm MDM are presented as afflicted by poor model simulation of their observed CO2. However no confidence is given about the JR-STATION sites in terms of the accuracy of model representation of CO2 at these sites, only a mean bias which is a very limited measure of modelâˇAˇTobservation fit improvement. This should be presented and extensively discussed prior to discussing the result of inverse modelling with a blind approach to the forward simulated CO2 (this is also directly related to the comment above).*

**Author's response**: Peters et al. (2010) mentioned "A second set of observations was deweighted because the large spread in model-minus-observed $CO_2$ suggested that our model regularly missed the model-data-mismatch target. This latter set includes the continuous data from the Cabauw tower (CBW0200_52C3), Westerland (WES_23C0), and Kasprowy Wierch (KAS_53C0) as well as the discrete samples from BAL_01D0, BSC_01D0, HUN_01D0, and OBN_01D0. The first two sites (CBW, WES) are known to be in highly industrialized regions and susceptible to strong fossil fuel burning influences and model representation error. For some of the other sites there was reason to doubt the representivity and/or data quality of parts of the time series.".

Therefore, we have added a reference (Peters et al., 2010) at the end of the texts in P11 l29.

According to reviewer's opinion, we have checked monthly averaged differences of the model $CO_2$ concentration and observed $CO_2$ concentration at JR sites in Fig_rev3. For the assimilation period, the JR experiment consistently shows smaller biases compared to the CNTL experiment, which implies that the model representation of $CO_2$ at JR sites is more accurate in the JR experiment than in the CNTL experiment. We have added the text to read, "In addition to the average bias for the entire period, the time series of monthly averaged bias of the model $CO_2$ concentration from the observed $CO_2$ concentration at JR-STATION sites shows that the JR experiment consistently shows smaller biases compared to the CNTL experiment (not shown), which implies that the model

representation of $CO_2$ at JR-STATION sites is more accurate in the JR experiment than in the CNTL experiment.".

[Figure]

Figure_rev 3. Time series of difference between observed and model $CO_2$ concentration simulated using optimized surface fluxes in CNTL experiment (blue), and JR experiment (red) at (a) AZV, (b) BRZ, (c) DEM, (d) IGR, (e) KRS, (f) NOY, (g) SVV, (h) VGN, and (i) YAK site. The differences are calculated by subtracting observed CO2 concentrations from model CO2 concentrations and averaged by month. Units are ppm.

*24) Overall section 3.2 is too limited and lacks a conclusion to support the subsequent analyses, especially in view of demonstrating the value of additional observations offered by JR-STATION.*

**Author's response**: We have revised Section 3.2 by including the comparison with the independent observations of vertical profiles which were not used for assimilation.

We have evaluated the posterior $CO_2$ concentrations from the two experiments with independent data. We used the airborne observations over BRZ (Berezorechka; 56.15°N, 84.33°E) in the taiga region of West Siberia (detailed explanation in Section 2.3) as the independent data. The results show that the optimized fluxes of JR experiment exhibit better agreement with independent observations in terms of root mean square difference, mean absolute error, and Pearson's correlation coefficient at all altitudes, which supports the usefulness of Siberian tower measurements on the estimation of surface $CO_2$ fluxes over Siberia. Table 5 and discussion of the results (Section 3.2) are added in the revised manuscript as follows.

Table 5. Bias, root mean square difference, mean absolute error, and Pearson's Correlation Coefficient of the model $CO_2$ concentration of CNTL and JR experiments in comparison with the vertical profile of $CO_2$ concentrations at BRZ site.

| Altitude (km) | Bias (ppm) | | Root-Mean-Square Difference (ppm) | | Mean Absolute Error (ppm) | | Pearson's Correlation Coefficient | |
|---|---|---|---|---|---|---|---|---|
| | CNTL | JR | CNTL | JR | CNTL | JR | CNTL | JR |
| ~ 0.5 | -0.13±4.81 | 0.20±4.57 | 4.82 | 4.57 | 3.45 | 3.23 | 0.95 | 0.95 |
| 0.5 ~ 1.0 | 0.58±4.30 | 0.83±4.10 | 4.34 | 4.18 | 3.14 | 3.03 | 0.95 | 0.95 |
| 1.0 ~ 1.5 | 0.40±3.94 | 0.56±3.69 | 3.96 | 3.74 | 2.88 | 2.68 | 0.93 | 0.94 |
| 1.5 ~ 2.0 | 0.25±3.46 | 0.42±3.24 | 3.47 | 3.27 | 2.49 | 2.34 | 0.93 | 0.94 |
| 2.0 ~ 2.5 | 0.43±3.20 | 0.59±2.91 | 3.22 | 2.97 | 2.35 | 2.18 | 0.92 | 0.94 |
| 2.5 ~ 3.0 | 0.56±2.89 | 0.73±2.58 | 2.94 | 2.69 | 2.21 | 2.08 | 0.90 | 0.92 |
| 3.0 ~ | 0.13±3.19 | 0.44±2.65 | 3.19 | 2.68 | 3.89 | 2.03 | 0.86 | 0.90 |

We have added the following sentences at the end of Section 3.2.

"In addition, model $CO_2$ concentrations calculated by optimized fluxes of the two experiments are compared with independent, not assimilated, vertical profiles of $CO_2$ concentration measurements by aircraft at BRZ site in Siberia. Table 5 presents the average bias, root-mean-square difference (RMSD), mean absolute error (MAE), and Pearson's correlation coefficient of the model $CO_2$ concentrations calculated by optimized fluxes of the two experiments based on the observations at BRZ site as the

reference. The statistics are calculated at each vertical bin with 500 meter interval. Overall, the biases of two experiments are less than 0.83 ppm showing good consistency between model and observed $CO_2$ concentrations. The biases of the CNTL experiment are smaller than those of the JR experiment at all altitudes, whereas the standard deviations of the CNTL experiment are greater than those of JR experiment, which implies that the biases of the CNTL experiment fluctuate as its average more than those of the JR experiment. In contrast, the RMSD and MAE of the JR experiment are smaller than those of the CNTL experiment, and the correlation coefficient of the JR experiment is greater than that of the CNTL experiments. Therefore, overall the statistics show that the model $CO_2$ concentrations of the JR experiment is relatively more consistent with independent $CO_2$ concentration observations compared to those of the CNTL experiment over Siberia."

*25) In section 3.3. Fig 7 8 and 9 are valuable but not sufficient in themselves to allow the reader to appreciate the contribution of the additional JR observations. A mapping of prior uncertainties, CNTL UR and JR UR would be required to support the discussion.*

**Author's response**: Following the reviewer's opinion, we have plotted the uncertainty reductions (UR) of CNTL experiment and JR experiment from their prior uncertainties and the difference of two URs (Fig_rev 4). A mapping of prior uncertainties is not shown because the prior uncertainties do not show the contribution of the additional observations. Except the EB region (i.e. Siberia), the average URs of two experiments show similar patterns in the Northern Hemisphere. The difference between the URs of CNTL (Fig._rev 4a) and JR (Fig._rev 4b) is readily apparent in Siberia (Fig._rev 4c), which is very similar result with the UR using Eq. (7) shown in Fig. 7c. Because the Fig. 7 in the manuscript already shows the contribution of the additional JR observations clearly and the URs of CNTL and JR from the prior uncertainties do not provide additional information on the impact of Siberian observations, we did not insert Fig._rev 4 in the revised manuscript. Instead, we have added the texts to read,

"The uncertainty reduction of CNTL and JR experiments based on the prior uncer tainty as the reference ($\sigma_{prior}$ used instead of $\sigma_{CNTL}$ in Eq. (8); $\sigma_{CNTL}$ or $\sigma_{JR}$ used instead of $\sigma_{JR}$ in Eq. (8)) shows similar values in the NH except in Siberia region (not shown). In addtion, the difference between average uncertainty reduction of C NTL and JR experiments based on the prior unceatinty as the reference (not shown) i s very similar to the average of uncertainty reduction in Eq. (8) shown in Fig.7a."

[Figure]

Figure_rev 4. Average uncertainty reduction (%), based on the prior uncertainty as a reference, of (a) CNTL experiment and (b) JR experiment. (c) The difference between (a) and (b).

*26) P12 l15: 'additional observations sometimes have a great impact' : Please be more explicit and quantitative about 'sometimes' and 'great impact'.*

**Author's response**: Following the reviewer's opinion in editorial comment 29, this statement and Fig. 7b were removed in the revised manuscript.

*27) P12 l16: The author find stronger UR in summer than in winter. Is this due to a higher uncertainty related to larger net fluxes in summer relative to winter? How is this seasonal UR difference explained over Siberia?*

**Authors's response**: As the reviewer's point, stronger uncertainty reduction of EB in summer than that in winter is due to a higher uncertainty related to larger net fluxes in

summer relative to winter. The correction of optimized surface $CO_2$ fluxes from prior fluxes is larger in summer than winter (Fig. 6a in revised manuscript). This is due to that the observations in Siberia exhibited large flux correction and uncertainty reduction in summer than winter.

Therefore, we have revised the texts to read, "The uncertainty reduction of EB in summer is higher than that in winter (Figs. 7b, c) due to a higher uncertainty associated with larger net fluxes in summer compared to winter (Fig. 6a).".

*28) P13 l7-8: please give more details about why high 1-week RMSD and self sensitivities of JR STATION sites is consistent.*

**Author's response**: The information content is a measure of the information extracted from the observations. The average information content is proportional to the average value of self-sensitivity and the number of observations used in the data assimilation. As shown in Kim et al. (2014b), the regions with large average information contents are consistent with the regions with a high RMSD, which implies that observations in that region provide much information. The JR-STATION tower sites have abundant observations and large self-sensitivities. Therefore, large self-sensitivities at JR-STATION is correlated with the large 1-week RMSD. To clarify, we have revised the texts to read, **"The RMSD of the analyzed surface $CO_2$ fluxes constrained by one week of observations from the background fluxes in JR experiment is greater than that in CNTL experiment (Figs. 9a, b), implying that surface $CO_2$ fluxes in Siberia are analyzed by JR-STATION data in Siberia directly at the first cycle. This is consistent with the high value of self-sensitivities at JR-STATION sites as shown in Fig. 8b. Because JR-STATION data are abundant and have large self-sensitivities, these observations provide large information on the estimated surface CO2 fluxes over Siberia in the first cycle. ~".

*29) P13 l9: 'it takes 5 weeks to affect the surface CO2 fluxes in Siberia by the transport of CO2 concentrations' : this statement is not supported by the demonstration in Kim et al. (2014, see their Fig 13), who only compared 1 week and 5 weeks, but not other time intervals. Therefore this incorrect interpretation needs to be reformulated. I could suggest a sentence such as 'it takes more than 1 week to affect the surface CO2. . .', which is better supported by the elements provided.*

**Author's response**: We have revised the texts as the reviewer suggested.

*30) However this observation by the authors is important. If the correction of Siberian surface fluxes, in CNTL, is only performed based on air masses between 1 and 5 weeks, it means that Siberia is an underconstrained region in terms of CO2 fluxes (and this is a*

*conservative statement). As a result, comparisons between CNTL and JR should take into account the large 'weakness' of fluxes allocated to Siberia by the CNTL inversion. This is illustrated in Table 2: EB is the region with the strongest increment between a priori and CNTL (from -0.07 to -1.17 PgC/y), but at the same time it is the only region with the least in situ observations. The other regions have smaller increment, and at the same time more 'local' observations are available. This bias needs to be explicated and discussed. The sentence of the abstract (p2, l1) comparing the fluxes calculated with the additional observations to the fluxes calculated without, suggests as such that the two flux values can be compared directly ('uptake. . . decreased'). On the contrary, Siberian fluxes calculated without the additional JR observations are highly sensitive to many assumptions within the inversion, and therefore any direct comparison requires a clear statement that this comparative approach is dependent on the inversion set up. This is also true for the sentence concluding this section (p13 l14-16).*

**Author's response**: Following the reviewer's opinion, we have added the texts to read, "The largest increment between a priori and CNTL is shown in EB with the least in situ observations as shown in Fig. 1. The other regions show smaller increment with more 'local' observations available." in Section 3.1.

The differences between the CNTL and JR are caused by additional JR-STATION data over Siberia, but not caused by the inversion set up because the two experiments use the same inversion modeling setup except JR-STATION data.

*31) Section 3.4. This section should also propose comparison with inversions intercomparison exercises, such as the TRANSCOM intercomparisons, see Gurney et al. (2002) or RECCAP Peylin et al (2013 Biogeosciences, 10, 6699-6720, doi:10.5194/bg-10-6699-2013). Please also compare with the synthesis work of Dolman et al. (2012)*

**Author's response**: Following the reviewer's opinion, the comparison with synthesis work of Dolamn et al. (2012) based on bottom-up and top-down method is included in Table 5 of the revised manuscript.

On the other hand, the other reviewer asked us to choose the same spatial domain (Eurasian Boreal and Europe) and temporal extent with other studies as similar as possible. The analysis period of mean fluxes in Gurney et al. (2002) is from 1992 to 1996 and that in Peylin et al. (2013) is from 2001 to 2004, which does not exactly match with the analysis period (2002-2009) in this study. Therefore, the results of Gurney et al. (2002) and Peylin et al. (2013) were not used in comparison with other studies.

We have revised the texts in Section 3.4 as follows.

"A comparison of the optimized surface $CO_2$ flux in this study with other previous studies is presented in Table 6. In the EB, the land sink from the JR experiment (-0.77±0.70 Pg C yr$^{-1}$) is smaller than those reported by Zhang et al. (2014b) (-1.02±0.91

Pg C yr$^{-1}$), Maki et al. (2010) (-1.46±0.41 Pg C yr$^{-1}$), and the CT2013B (CarbonTracker released on 9 Feburary 2015; documented online at http://www.esrl.noaa.gov/gmd/ccgg/carbontracker/CT2013B/) results (-1.00±3.75 Pg C yr$^{-1}$), but higher than those reported by Saeki et al. (2013) (-0.35±0.61 Pg C yr$^{-1}$; including biomass burning 0.11 Pg C yr$^{-1}$)), and similar with those reported by Dolman et al. (2012) (-0.613 Pg C yr$^{-1}$).

Because CT2013B and Zhang et al. (2014b) use the similar inversion framework as this study, the reduced land sink is caused by assimilating additional observations. The difference in land sink between the JR experiment and Saeki et al. (2013) is caused by a different inversion system framework which includes prior flux information, atmospheric transport model, observation data set, and inversion method. Despite different inversion system framework used in each study, two studies using the JR-STAITON data exhibit similar results in relative terms, reduced uptake of $CO_2$ fluxes and uncertainties over Siberia. Nontherless, the land sink from the JR experiment is somewhat different with other inversion results, its value falls within the flux uncertainty range. Although the land sink in Dolamn et al. (2012) is the average land sink obtained from three methods (inventory-based, eddy covariance, and inversion methods) and estimated not only for Siberia but for Russian territory including Ukraine, Belarus, and Kazakhstan, the land sinks of the JR experiment and Dolman et al. (2012) shows similar values. Overall, the optimized surface $CO_2$ fluxes in EB of JR experiment are comparable to those of other previous studies."

*32) P13 l26 the paragraph is concluded by the importance of the inversion framework used. Therefore the difference with the results of Saeki et al. (2013) needs to be examined in more detail. This paragraph leads the reader to the obvious conclusion that the choice of the inversion setup has as much impact on the posterior fluxes uncertainty of Siberia than the addition of a novel observation network such as JR-STATION.*

*Comparing the study reviewed and Saeki et al. (2013) from the numbers provided in the papers, it is striking that the two studies consistently conclude, in relative terms, both to a Siberian flux that is lower by about one third when adding JR-STATION (-37.5% for Saeki et al., -34.1% for this study), and decrease their uncertainty by about one-quarter (-22.7% and -24.7% respectively). At the same times, when comparing the two studies (being based on similar observation data sets), Kim et al.'s find fluxes consistently two times higher that Saeki et al. with or without the JR-STATION dataset, and report uncertainty that are 15% higher, with or without the JR STATION dataset.*

*Therefore the reported numbers lead to the observations that a change in inversion setup has more impact on the estimated fluxes in Siberia than using or not the only existing dataset in the region. This is stimulating because it means that more research is needed before CO2 budgets calculated using inverse methods over Siberia can be reliably used. It supports the suggestion for extensive comparison of the simulated and measured CO2 at the JR STATION*

*sites in this study.*

*The numbers above also imply that the current set up questions the constraints on CO2 fluxes (in terms of range of likely flux value) over Siberia reported by Saeki et al. 2013, and this requires further comparison with other studies, possibly bottom up allometric or modelling studies. It should be noted also that Maki et al. 2010 reported smaller uncertainty (0.41 PgC/y), even without the JR-STATION network. This requires a detailed comparison of these studies for the sake of coherence.*

**Author's response**: Although both this study and Saeki et al. (2013) used JR station data, prior flux information (e.g. biosphere, ocean, fires, and fossil fuel fluxes), atmospheric transport model, observation data set, and inversion method of two studies are different. Therefore, the different flux values in EB in this study and Saeki et al. (2013) are caused by not only the inversion setup but also most components which constitute the inversion system framework. The term "inversion system framework" used in P3 l26 denotes prior flux information (e.g. biosphere, ocean, fires, and fossil fuel fluxes), atmospheric transport model, observation data set, and inversion method.

Despite several differences in the inversion system frameworks, the optimized surface $CO_2$ fluxes over Siberia in this study and Saeki et al. (2013) show the similar conclusions in relative terms but different magnitudes in terms of fluxes and their uncertainties. These discrepancies of estimated surface fluxes over specific regions among inversion systems are already reported in previous studies. For example, in the intercomparison study using 11 inversion systems, Peylin et al. (2013) demonstrated that: (1) there are more consistencies between inversions for larger scales such as interannual variability of global fluxes; (2) the largest total land sink in the Northern Hemisphere is nearly unanimously located in the Eurasian domain, whereas a large spread among the inversions remains for the specific regions (e.g., North America, Europe, North Asia) in Northern Hemisphere. Consequently, the longitudinal breakdown of the total northern sink appears to be much more variable than the total flux itself. These characteristics are also shown in the comparison between this study and Saeki et al. (2013). The average global total flux is similar between this study (-3.61±1.73 Pg C yr$^{-1}$ ) and Saeki et al. (2013) (-3.51±3.18 Pg C yr$^{-1}$). However, the partition between land and ocean fluxes and fluxes in EB region are different between two studies (note the table below for more information).

| Region | This study (2002-2009) | | Saeki et al. (2013) (2000-2009) | |
|---|---|---|---|---|
| | Without Siberian data | With Siberian data | Without Siberian data | With Siberian data |
| Global total | -3.60±1.85 | -3.61±1.73 | -3.50±3.26 | -3.51±3.18 |
| Global land | -1.68±1.57 | -1.61±1.43 | -1.95±3.08 | -1.90±3.00 |
| Global ocean | -1.91±0.97 | -2.00±0.97 | -1.55±1.06 | -1.61±1.06 |
| Eurasian Boreal | -1.04±0.93 | -0.64±0.70 | -0.56±0.79 | -0.35±0.61 |

*For the land and ocean fluxes, biomass-burning (fire) emissions are included and fossil

fuel emissions are not included.

Anyway, flowing the reviewer's opinion, we have added the comparison with other study based on bottom-up and modelling studies (Note detailed comment 31).

The object of this study is to compare between optimized surface $CO_2$ fluxes in CNTL experiment and those in JR experiment within the same inversion system. In this sense, verification of the inversion results using the independent observations would be beneficial. As shown in Fig_rev2, during the assimilation period, the JR experiment consistently shows smaller biases compared to the CNTL experiment, which implies that the model representation of $CO_2$ at JR sites is more accurate in the JR experiment than in the CNTL experiment. In addition, the root-mean-square difference (RMSD) and mean absolute error (MAE) of model $CO_2$ concentration calculated by optimized fluxes in JR experiment exhibits better agreement with independent observation which is not used in the optimization (vertical profile of $CO_2$ concentration measured over BRZ site).

Thus, we have revised the text to reads, "The difference in land sink between the JR experiment and Saeki et al. (2013) is caused by a different inversion system framework which includes prior flux information, atmospheric transport model, observation data set, and inversion method. Despite different inversion system framework used in each study, two studies using the JR-STAITON data exhibit similar results in relative terms, reduced uptake of $CO_2$ fluxes and uncertainties over Siberia."

*33) P14 l2 and 3: what is the uncertainty range of single-year flux values (here 2008 and 2009 are discussed)?*

**Author's response**: Following the reviewer's opinion, we have revised the manuscript as follows.

"The land sinks of the JR experiment in 2008 and 2009 are -0.73±0.41 and -0.76±0.38 Pg C yr$^{-1}$, respectively, whereas much lower uptakes (-0.21±0.49, -0.38±0.43 Pg C yr$^{-1}$) are obtained for the CNTL experiment."

*34) Section 4 Please revise the conclusions according to the discussion above. What are the key challenges identified by the authors before estimating robust $CO_2$ fluxes over Siberia from atmospheric inverse modelling?*

**Author's response**: Following the reviewer's opinion, we have revised the conclusion section as follows. The key challenge is overcoming the sparseness of atmospheric $CO_2$ observing network over Siberia to better estimate the surface $CO_2$ fluxes over Siberia.

"The global balances of the sources and sinks of surface $CO_2$ fluxes were maintained with a similar trend for both experiments, while the distribution of the optimized surface

CO₂ fluxes changed. The magnitude of the optimized biosphere surface $CO_2$ uptake and its uncertainty in EB (Siberia) was decreased from $-1.17\pm0.93$ Pg C yr$^{-1}$ to $-0.77\pm0.70$ Pg C yr$^{-1}$, whereas it was increased in other regions of the NH (Eurasian Temperate, Europe, North American Boreal, and North American Temperate). The land sink of Europe increased significantly for 2008 and 2009, which is consistent with the other inversion results inferred by satellite observations. Additional observations are used to correct the surface $CO_2$ uptake in June and July, the active vegetation uptake season, in terms of monthly average optimized surface $CO_2$ fluxes. As a result, the additional observations do not exhibit a change in the magnitude of the global surface $CO_2$ flux balance because they provide detailed information about the Siberian land sink instead of the global land sink magnitude, when they are used in the well-constructed inversion modeling system.

The model $CO_2$ concentration using the background and optimized surface $CO_2$ fluxes in the JR experiment are more consistent with the $CO_2$ observations used in the optimization than those in the CNTL experiment, showing lower biases in the EB region. In contrast, the differences of biases in ET and Europe between the two experiments are not distinguishable. In comparison with vertical profiles of $CO_2$ concentration observations which are not used in the optimization, the model $CO_2$ concentrations in the JR experiment show the smaller RMSD and MAE, and the greater correlation coefficient that those in CNTL experiment.

The new observations provide useful information on the optimized surface $CO_2$ fluxes. The observation impact of the Siberian observation data is investigated by means of uncertainty reduction and self-sensitivity calculated by an influence matrix. Additional observations reduce the uncertainty of the optimized surface $CO_2$ fluxes in Asia and Europe, mainly in the EB (Siberia), where the new observations are used in the assimilation. The average self-sensitivities of the JR-STATION sites are approximately 60% larger than those at other continuous measurements (e.g., tower measurements in North America). The global average self-sensitivity and cumulative impact of the JR experiment are higher than that of the CNTL experiment, which implies that the individual observation impact of JR-STATION data on optimized surface $CO_2$ fluxes is higher than the average values. The RMSD of the analyzed surface $CO_2$ fluxes constrained by one week of observations from the background fluxes also suggests that new Siberian observations provide a larger amount of information on the optimized surface $CO_2$ fluxes.”

*35) P15 l9 is the longitudinal redistribution toward Europe a conclusion shared by Saeki et al., 2013?*

**Author's reponse**: Saeki et al. (2013) didn't show the longitudinal redistribution toward Europe. This is a finding of this study. Thus, we have revised the misleading text as follows.

"This study shows that the JR-STATION data affect the longitudinal distribution of the total NH sinks, especially in the EB and Europe, when it is used by atmospheric $CO_2$ inversion modeling."

*36) P32 fig 5 further discussion of the trend in European fluxes and its possible drivers would be valuable.*

**Author's response**: Following the reviewer's opinion, we have added further discussions on the trends in European fluxes as follows.

"Figure 5 is the same as Fig. 4 but covers land regions in the NH. Although the optimized surface $CO_2$ fluxes over global total are similar, those over each TransCom region are different in each experiment. The optimized fluxes over each region show greater annual uptake relative to the prior fluxes in both experiment. The difference between the two experiments is largest in the EB as expected (Fig. 5a). The JR experiment exhibits a weaker surface $CO_2$ uptake in the EB than does the CNTL experiment except for 2003 as shown in Fig. 3b, whereas the JR experiment exhibits a greater surface $CO_2$ uptake in the other regions, especially over Europe in 2008 and 2009, than the CNTL experiment (Figs. 5b, c, d, and e). It is driven by the increase of $CO_2$ uptake in Eastern Europe (Figs. 3g and h). Because most of JR-STATION sites are located in the western part of Siberia (Fig. 1), the optimized surface $CO_2$ fluxes over Eastern Europe could be affected by JR-STATION observations. As a result, the trends of the surface $CO_2$ uptake of EB and Europe in two experiments are different. The trend of EB in CNTL experiment is -0.06 Pg C $yr^{-2}$, whereas that in JR experiment is 0.02 Pg C $yr^{-2}$ due to the reduced uptake of $CO_2$ in JR experiment since 2005 (Fig 5a). In contrast, the surface $CO_2$ uptake trends of other land regions in NH are similar between two experiments."

*37) P33 fig 6 please add the prior fluxes seasonal cycles to each panel*

**Author's response**: We have added prior fluxes seasonal cycles in Fig. 6 following the reviewer's opinion.

**Editorial Comments:**

*1) There are several occurrences of unexplained acronyms; the authors should check this carefully.*

    **Author's response**: We have explained acronyms in the revised manuscript.

*2) P2 l13: typo: Schulze et al, not Schuleze et al*

    **Author's response**: We have corrected the typo.

*3) P3 l10: 'useful information': please be more specific on the usefulness of this information.*

    **Author's response**: Following the reviewer's opinion, we have revised the texts to read, "Though a broad spatial coverage of $XCO_2$ from satellite radiance observations provides useful information for inversion systems in quantifying surface $CO_2$ fluxes at various scales which is not provided by ground-based measurements, the current $XCO_2$ has low accuracy and regional biases of a few tenths of a ppm, which may hamper the accuracy of estimated surface $CO_2$ fluxes (Miller et al., 2007; Chevallier et al., 2007)."

*4) P3 l18: due to the difference in time period the word 'accompany' is not correct, should be e.g. 'preceded'*

    **Author's response**: We have revised the word following the reviewer's opinion.

*5) P3 l21: typo: YAK-AEROBO -> YAK-AEROSIB.*

    **Author's response**: We have corrected the typo.

*6) P3 l25 : multi-year Zotto measurements can certainly be used for inverse modelling.*

    **Author's response:** Following the reviewer's opinion, we have revised the texts as follows.

    "However, except Zotino that has multi-year measurements, these data collected during specific seasons or over only a few years do not provide the long-term $CO_2$ concentration data necessary to be used as a constraint in the inverse modeling system."

*7) P4 l11: 'increasing the sites': should be 'increasing the number of sites'*

**Author's response**: We have revised the texts following the reviewer's opinion.

*8) P5 l19: expand EnKF acronym*

**Author's response**: We have expanded the acronym.

*9) P5 l22 (eq. 2) and l26: please be consistent on notation: x (superscript) b or x (subscript)*

**Author's response**: We have revised the notation.

*10) P6 l7-13: ensembles, perturbation, background error, localization, physical distance: these notions have not been introduced before, please provide guidance for the reader to understand*

**Author's response:** Following the reviewer's opinion, we have provided references as follows.

"The detailed algorithm of inversion method used in this study can be found in Peters et al. (2007) and Kim et al. (2014a). "

*11) P6 l15-17: please revise this sentence for clarity and syntax*

**Author's response:** Following the reviewer's opinion, we have revised the manuscript as follows.

"Statistical significance test is performed on the linear correlation coefficient with a cut-off at a 95% significance in a student's T-test. Then the components of Kalman gain with an insignificant statistical value are set to zero."

*12) P7 l7 please give references for CDIAC and EDGAR.*

**Author's response:** Following the reviewer's opinion, we have provided the references.

*13) P7 l12: please provide link for ESRL data set, and give credit consistently to organization operating the measurements (e.g. in Europe).*

**Author's response:** Following the reviewer's opinion, we have provided link for ESRL

data set as follows.

"(1) surface $CO_2$ observations distributed by the NOAA ESRL (observation sites operated by NOAA, Environment Canada (EC), the Australian Commonwealth Scientific and Industrial Research Organization (CSIRO), the National Center for Atmospheric Research (NCAR), Lawrence Berkeley National Laboratory (LBNL)) (observation data is available at http://www.esrl.noaa.gov/gmd/ccgg/obspack/data.php; Masarie et al., 2014) "

In addition, the organizations operating the measurements are denoted in Table 1 and credited in Acknowledgments.

*14) P7 l29: Is the MDM 'determined' or incremented? It is unclear with this formulation. Please revise accordingly. What is intended by 'innovation' chi-2?*

**Author's response**: The MDM is determined and innovation is $y^o$-$\mathbf{H}x^b$. Therefore innovation $\chi^2$ statistics is $\chi^2$ formulation in Eq. (7). To clarify, we have revised the texts indicated by the reviewer as follows.

"In CarbonTracker, model data mismatch (MDM, **R** in Eq. (7)) is assigned by site categories. The location of each observation site is represented in Fig. 1. The assigned MDM requires innovation $\chi^2$ statistics in Eq. (7) become close to one at each observation site (Peters et al. 2007).

$$\chi^2 = \frac{(y^o\text{-}\mathbf{H}x^b)^2}{\mathbf{HP}^b\mathbf{H}^T\text{+}\mathbf{R}} ,$$
(7),

where $y^o$-$\mathbf{H}x^b$ represent innovation."

*15) P8 l17: typo: exists -> exist*

**Author's response**: We have corrected the typo.

*16) P8 l25: 'from two experiments': I suggest to change to a determined form 'from the two experiments'*

**Author's response**: We have revised the texts as the reviewer suggested.

*17) P9 l4 'greater': please quantify in the text*

**Author's response**: Following the reviewer's opinion, we have revised the texts to read,

"The optimized biosphere flux uptakes of the CNTL and JR experiments are globally 1.60 ~ 1.61 Pg C yr$^{-1}$ greater than the prior flux uptakes (Figs. 2a, c, d, Table 2).".

*18) P9 l14: global total optimized CO2 fluxes': the wording is problematic because this does not include fossil fuel and forest fire. Should be e.g. 'total biogenic'.*

**Author's response**: Because these fluxes are sum of biosphere and ocean fluxes (fossil fuel and forest fire are not included) over the globe, we have revised the text to read, "The global total biogenic and oceanic optimized CO2 fluxes are ~"

*19) P9 l20: typo 'between two the experiments' -> 'between the two experiments'*

**Author's response**: We have corrected the typo.

*20) P10 l2 please revise and clarify: 'is reduced all years' , I suggest 'is reduced in JR for all years'*

**Author's response**: We have revised the texts as the reviewer suggested.

*21) P10 l9 readability would be improve to write Siberia instead of 'EB'. What is 'ET'?*

**Author's response**: We have revised the texts as the reviewer suggested.

*22) P10 l14 the figure is not a histogram (binned distribution) but a time series. Please correct accordingly.*

**Author's response**: We have corrected the mistake.

*23) P11 l1 'Additional Siberian data": very indeterminate formulation. Please add e.g. 'These additional JR Siberian data'*

**Author's response**: We have revised the texts as the reviewer suggested.

*24) P11 l17 what are background scaling factor? This seemingly important notion should be*

*explained in the section on experimental framework.*

**Author's response**: We have revised the Section 2.1 as follows.

"After one analysis step is completed, the new mean scaling factor that serves as the background scaling factor for next analysis cycle is predicted as

$$\lambda_t^b = \frac{(\lambda_{t-2}^a + \lambda_{t-1}^a + 1)}{3}, \tag{6}$$

where $\lambda_t^b$ is a prior mean scaling factor of the current analysis cycle, $\lambda_{t-2}^a$ and $\lambda_{t-1}^a$ are posterior mean scaling factors of previous cycles. Eq. (6) propagates information from one step to the next step (Peters et al., 2007)"

*25) P11 l22 ET: explain acronym*

**Author's response**: To answer the question 21) above, we have explained ET earlier in the manuscript.

*26) Section 3.3. : Sect. 3.3. or part thereof should be before 3.1 and 3.2 as these sections 3.1 and 3.2 discuss already on the basis of CNTL vs JR runs. The structure of the paper could be reassessed for the benefit of readability.*

**Author's response**: As the reviewer indicated, the Sections 3.1 and 3.2 discuss already the difference between CNTL and JR runs in terms of carbon fluxes and concentrations. Section 3.3 discusses the difference between CNTL and JR in terms of uncertainty reduction and observation impact on data assimilation. We think the differences on fluxes and concentrations need to be discussed first, and the uncertainty reduction and observation impact can follow. The original title of Section 3.3 does not represent what it deals with appropriately. Therefore, instead of changing the order of Sections, we have retitled the Section 3.3 as 'uncertainty reduction and observation impact' for better readability.

*27) P12 l10: Conifer: typo (confer). Why are only Conifer Forest of EB mentioned with no discussion of other ecosystems? What do they represent vs the rest of the Siberian ecosystems?*

**Author's response**: We have corrected the typo. We only discussed the Conifer Forest of EB because JR stations are mainly located in the Conifer Forest of EB. In addition, ecoregions close to the Conifer Forest of EB show relatively large value of uncertainty reduction as mentioned.

*28) P12 l11 'which has additional information'; I suggest to use a determinate form 'which has the additional information'*

**Author's response**: We have revised the texts as the reviewer suggested.

*29) P12 l14: 'the magnitude of the maximum uncertainty reduction is higher than the average value': this is certainly trivial. Please remove this sentence. I don't see the value of maximum weekly UR at all and I suggest to remove this panel 7b*

**Author's response**: Following the reviewer's opinion, we have removed that sentence and Fig. 7b.

*30) P12 l26 please relate the definition of self sensitivity to the*

**Author's response**: Although, it is hard to recognize what the reviewer intended in this comment, we have revised the manuscript as follows.

"The self-sensitivity is the diagonal element of the influence matrix which measures the impact of individual observations in the observation space on the optimized surface $CO_2$ flux. The large self-sensitivity value implies that the information extracted from observations is large. Figure 8 shows the self-sensitivities of the two experiments averaged from 2002 to 2009. The average self-sensitivities at the JR-STATION sites are approximately 60% larger than those at the towers in North America, i.e., Continuous site category observations in Fig. 1."

*31) P12 l26 what are 'Continuous site category observations'?*

**Author's response**: The observation sites used in CarbonTracker are categorized as marine boundary layer, continental sites, mixed land/ocean and mountain sites, continuous sites, and difficult sites. Continuous site category observations are observations sampled continuously. JR-STATION observations are sampled continuously, thus assigned to continuous site category. To clarify, we have added following texts in the last paragraph of Section 2.3.

"The site categories and MDM values are assigned the same value as in previous studies (Peters et al., 2007; Kim et al. 2014b; Zhang et al., 2014b): marine boundary layer (0.75 ppm), continental sites (2.5 ppm), mixed land/ocean and mountain sites (1.5 ppm), continuous sites (3.0 ppm), and difficult sites (7.5 ppm). Continuous site category is generally used for observations measured continuously. For the JR-STATION sites that have continuous tower measurements, the MDM is set to 3 ppm, which is the same as for

tower measurements in North America."

*32) P13 l6-7: 'fluxes. . . are analysed by direct observations at the first cycle': this sentence might not be clear. Please rephrase.*

**Author's response**: Following the reviewer's opinion, we have revised the manuscript as follows.

"The RMSD of the analyzed surface $CO_2$ fluxes constrained by one week of observations from the background fluxes in JR experiment is greater than that in CNTL experiment (Figs. 9a, b), implying that surface $CO_2$ fluxes in Siberia are analyzed by JR-STATION data in Siberia directly at the first cycle."

*33) P13 l21 What is CT2013B?*

**Author's response**: CT2013B is the CarbonTraker released by NOAA on 9 Feburary 2015. We have revised the manuscript as follows.

"In the EB, the land sink from the JR experiment (-0.77±0.70 Pg C $yr^{-1}$) is smaller than those reported by Zhang et al. (2014b) (-1.02±0.91 Pg C $yr^{-1}$), Maki et al. (2010) (-1.46±0.41 Pg C $yr^{-1}$), and the CT2013B (CarbonTracker released on 9 Feburary 2015; documented online at http://www.esrl.noaa.gov/gmd/ccgg/carbontracker/CT2013B/) results (-1.00±3.75 Pg C $yr^{-1}$), but higher than those reported by Saeki et al. (2013) (-0.35±0.61 Pg C $yr^{-1}$; including biomass burning 0.11 Pg C $yr^{-1}$) and Dolman et al. (2012) (-0.613 Pg C $yr^{-1}$). "

*34) P13 l22 Please be careful when reporting numbers. The uncertainty of 0.41 PgC/y is wrong , the correct number is given in your Table 5 (0.61).*

**Author's response**: We have corrected the typo.

*35) P15 l20. Please add acknowledgments for other observational data providers.*

**Author's response**: Following the reviewer's opinion, we have revised the acknowledgments as follows.

"The authors appreciate two reviewers for their valuable comments. This study was funded by the Korea Meteorological Administration Research and Development Program under the Grant KMIPA 2015-2021. The JR-STATION is supported by the Global

Environment Research Account for National Institutes of the Ministry of the Environment, Japan and the Russian Foundation for Basic Research (Grant No. 14-05-00590). The authors also acknowledge atmospheric $CO_2$ measurements data providers and cooperating agencies at China Meteorological Administration, Commonwealth Scientific and Industrial Research Organization, Environment Canada, Finnish Meteorological Institute, Hungarian Meteorological Service, Japan Meteorological Agency, Lawrence Berkeley National Laboratory, National Institute of Environmental Research, Norwegian Meteorological Institute, Max Planck Institute for Biogeochemistry, Morski Instytut Rybacki, National Center for Atmospheric Research, National Oceanic and Atmospheric Administration Earth System Research Laboratory, and Romanian Marine Research Institute."

*36) P23 Table 1 the table needs to differentiate altitude and sampling height, which is a potential indicator of how difficult it is to simulate a particular site. Also please make sure the proper credits are given to the providing Laboratories (last 7 lines, in Europe).*

**Author's response**: Following the reviewer's opinion, we have revised Table 1. For sites which have different altitude and sampling height, the sampling height is added. In addition, we have given the proper credits to the providing laboratories.

*37) P24 Table 2 I suggest to add total Northern hemisphere*

**Author's response**: Following the reviewer's opinion, we have added surface fluxes of total Northern hemisphere. In addition, Tropical total and Southern Hemisphere total are also added in Table 2.

*38) P27 CT2013B and CTE2014: please give reference.*

**Author's response**: Following the reviewer's suggestion, we have revised Table 6 to include references of CT2013B and CTE2014.

*39) P32 Fig 5 Panel (a): please correct typo (Euraisan -> Eurasian)*

**Author's response**: We have corrected the typo.

*40) P32 l3 there are no 'ocean' in this figure, please correct Fig 5 caption accordingly.*

**Author's response**: We have revised the caption of Fig. 5.

**References**

[revised manuscript text omitted]

---

## Author Response (AR2)

ACP-2015-875 (Editor – William Lahoz)

Response to Reviewer

The authors thank the reviewer for a thoughtful review of the manuscript. The responses for the reviewer's specific comments are as follows.

**General Comment:**

*The revised manuscript is a significant improvement relative to the original version – both in terms of scientific content and overall readability. The authors have done a credible job in addressing most of the reviewers' concerns, especially with the addition of Section 3.2 and Table 6. I have one major comment, which the authors can address with a short discussion (and/or figure). A few other minor typographical errors need to be corrected. I recommend the manuscript for publication in ACP.*

   **Author's response**: Following the reviewer's suggestions, we have revised the manuscript.

**Major Comments:**

*The fact that the posterior CO2 concentrations from the JR experiment shows a larger bias (relative to the CNTL experiment) when compared with the independent aircraft observations is disconcerting. Even though the RMSD and the MAE are lower for the JR experiment, the positive bias indicates that the JR inverse modeling setup generates more CO2 concentrations than the CNTL experiment. Based on mass balance, it is not surprising then that the JR experiment shows a reduced uptake in the region. It is not clear over what time period the statistics have been calculated for (also see Minor Comment #6). Hence a little more detail may be beneficial here. Have the authors investigated this bias issue in more detail? It may also be useful to add an additional figure showing the vertical profiles from the aircraft for one or two specific flights, and the corresponding posterior CO2 concentrations from the two inverse modeling experiments.*

   **Author's response**: The aircraft observations used in the verification are available over the similar period (2002-2009) as the JR-STATION data. The frequency of the aircraft flights is generally two to four times per month. The aircraft measurements were conducted in the afternoon on good weather days (Sasakawa et al., 2013). The statistics were calculated by using all of aircraft measurement.

   Following the reviewer's opinion, we investigated bias issue in more detail. We recalculated statistics by using aircraft measurement observed between 1200-1600 LST. This is the same time period applied to the daytime averaged $CO_2$ concentration of surface

measurements used in the assimilation. Near the surface, the result of JR experiment is better than that of CNTL experiment in terms of bias. The bias of the JR experiment is smaller than that of the CNTL experiment at the level under 500 m, whereas the biases of the CNTL experiment are smaller than those of the JR experiment at the levels above 500m. As the reviewer's point, JR experiment generates more $CO_2$ concentration over Siberia by reduced uptake of surface $CO_2$ fluxes. Other statistics (RMSE and MAE) at all altitudes of JR experiment are still less than that those of CNTL experiment after recalculation.

We revised the Table 5 as follows.

Table 5. Bias, root mean square difference, mean absolute error, and Pearson's Correlation Coefficient of the model $CO_2$ concentration of CNTL and JR experiments in comparison with the vertical profile of $CO_2$ concentrations at BRZ site.

| Altitude (km) | Bias (ppm) | | Root-Mean-Square Difference (ppm) | | Mean Absolute Error (ppm) | | Pearson's Correlation Coefficient | |
|---|---|---|---|---|---|---|---|---|
| | CNTL | JR | CNTL | JR | CNTL | JR | CNTL | JR |
| ~ 0.5 | -0.38±4.73 | -0.05±4.39 | 4.06 | 3.75 | 3.42 | 3.07 | 0.94 | 0.95 |
| 0.5 ~ 1.0 | 0.23±4.05 | 0.42±3.75 | 3.58 | 3.33 | 2.94 | 2.72 | 0.94 | 0.95 |
| 1.0 ~ 1.5 | 0.19±3.80 | 0.31±3.53 | 3.35 | 3.11 | 2.70 | 2.49 | 0.94 | 0.95 |
| 1.5 ~ 2.0 | 0.22±3.38 | 0.33±3.19 | 2.94 | 2.79 | 2.33 | 2.19 | 0.93 | 0.94 |
| 2.0 ~ 2.5 | 0.02±3.19 | 0.08±3.07 | 2.64 | 2.54 | 2.19 | 2.11 | 0.93 | 0.94 |
| 2.5 ~ 3.0 | 0.79±2.84 | 0.80±2.53 | 1.44 | 1.30 | 2.21 | 1.99 | 0.92 | 0.94 |
| 3.0 ~ | 0.61±3.15 | 0.61±2.91 | 1.49 | 1.38 | 2.42 | 2.26 | 0.88 | 0.91 |

We have revised the last paragraph of Section 3.2 as follows. The underlined parts denote added or revised sentences.

"In addition, model $CO_2$ concentrations calculated by optimized fluxes of the two experiments are compared with independent, not assimilated, vertical profiles of $CO_2$ concentration measurements by aircraft at BRZ site in Siberia. Aircraft measurements were conducted in the afternoon on good weather days. The frequency of flight was usually two to four times per month (Sasakawa et el., 2013). Table 5 presents the average bias, root-mean-square difference (RMSD), mean absolute error (MAE), and Pearson's correlation coefficient of the model CO2 concentrations calculated by optimized fluxes of the two experiments based on the observations at BRZ site as the reference. The statistics are calculated at each vertical bin with 500 meter interval by using aircraft measurements observed between 1200 – 1600 LST. Overall, the biases of two experiments are less than 0.80 ppm showing good consistency between model and observed $CO_2$ concentrations. Near the surface, the result of JR experiment is better than that of CNTL experiment in terms of bias. The bias of the JR experiment is smaller than those of the CNTL experiment

at the level under 500 m, whereas the biases of the CNTL experiment are smaller than those of the JR experiment at the levels above 500 m. The more $CO_2$ concentrations are generated over BRZ site because of the reduced uptake of surface $CO_2$ fluxes over Siberia in JR experiment. The standard deviations of the CNTL experiment are greater than those of JR experiment, which implies that the biases of the CNTL experiment fluctuate as its average more than those of the JR experiment. In contrast, the RMSD and MAE of the JR experiment are smaller than those of the CNTL experiment, and the correlation coefficient of the JR experiment is greater than that of the CNTL experiments. Therefore, overall the statistics show that the model $CO_2$ concentrations of the JR experiment is relatively more consistent with independent $CO_2$ concentration observations compared to those of the CNTL experiment over Siberia."

**Minor Comments:**

*1) Page 3, Line 17: Change 'column-averaged model' to 'column-average mole'*

**Author's response**: We have revised the text as the reviewer suggested.

*2) Page 3, Line 19: The word 'in situ' is irrelevant here*

**Author's response**: We have revised the text as the reviewer suggested.

*3) Page 3, Line 29: Check the spelling of Max Planck Institute*

**Author's response**: We have corrected the typo.

*4) Page 4, Line 28: Substitute the phrase 'dependence on' with 'sensitivity to'*

**Author's response**: We have revised the text as the reviewer suggested.

*5) Page 7, Lines 2 - 3: This sentence should be rephrased to increase its readability.*

**Author's response**: Following the reviewer's opinion, we have revised the text as follows.

"The sampling error caused by the limited ensemble size may degrade the analysis accuracy. To reduce the impact of sampling error in the EnKF, the covariance localization method is used"

*6) Page 14, Line 10-11: Are the aircraft observations available over the same period as the surface network? At what frequency are the aircraft flights carried out? And over what time period are the statistics calculated?*

**Author's response**: The aircraft observations used in the verification are available over the similar period (2002-2009) as the JR-STATION data. The frequency of the aircraft flights is generally two to four times per month. The aircraft measurements were conducted in the afternoon on good weather days (Sasakawa et al., 2013). The statistics were calculated by using all of aircraft measurement.

*7) Page 15, Line 18: Check the spelling of 'uncertainty'*

**Author's response**: We have corrected the typo.

*8) Page 17, Line 3: Check the spelling of 'JR-STATION'*

**Author's response**: We have corrected the typo.

*9) Page 17, Line 6: Check the spelling of 'Dolman et al.'*

**Author's response**: We have corrected the typo.

**Reference**

Sasakawa, M., Machida, T., Tsuda, N., Arshinov, M., Davydov, D., Fofonov, A., and Krasnov, O.: Aircraft and tower measurements of CO2 concentration in the planetary boundary layer and the lower free troposphere over southern taiga in West Siberia: Long-term records from 2002 to 2011, J. Geophys. Res. Atmos., 118, 9489-9498, doi:10.1002/jgrd.50755, 2013.